# Ependymal cell lineage reprogramming as a potential therapeutic intervention for hydrocephalus

Konstantina Kaplani[1], Maria-Eleni Lalioti[1], Styliani Vassalou [1], Georgia Lokka[1], Evangelia Parlapani[1], Georgios Kritikos[1], Zoi Lygerou [2] & Stavros Taraviras [1]✉

## Abstract

Hydrocephalus is a common neurological condition, characterized by the excessive accumulation of cerebrospinal fluid in the cerebral ventricles. Primary treatments for hydrocephalus mainly involve neurosurgical cerebrospinal fluid diversion, which hold high morbidity and failure rates, highlighting the necessity for the discovery of novel therapeutic approaches. Although the pathophysiology of hydrocephalus is highly multifactorial, impaired function of the brain ependymal cells plays a fundamental role in hydrocephalus. Here we show that GemC1 and McIdas, key regulators of multiciliated ependymal cell fate determination, induce direct cellular reprogramming towards ependyma. Our study reveals that ectopic expression of GemC1 and McIdas reprograms cortical astrocytes and programs mouse embryonic stem cells into ependyma. McIdas is sufficient to establish functional activity in the reprogrammed astrocytes. Furthermore, we show that McIdas' expression promotes ependymal cell regeneration in two different postnatal hydrocephalus mouse models: an intracranial hemorrhage and a genetic form of hydrocephalus and ameliorates the cytoarchitecture of the neurogenic niche. Our study provides evidence on the restoration of ependyma in animal models mimicking hydrocephalus that could be exploited towards future therapeutic interventions.

**Keywords** Ependymal Cells; Reprogramming; Hydrocephalus; McIdas; GemC1
**Subject Categories** Neuroscience; Stem Cells & Regenerative Medicine

## Introduction

Hydrocephalus is a neurological disorder defined by the abnormal accumulation of cerebrospinal fluid (CSF) within the brain ventricles, leading to increased head circumference (Schrander-Stumpel and Fryns, 1998). The condition can arise as a primary clinical feature triggered by a genetic cause (congenital hydrocephalus) or secondary (acquired hydrocephalus) to other insults such as central nervous system (CNS) infections, tumors, trauma, or hemorrhage (Rekate, 2009; Kahle et al, 2016; Kousi and Katsanis, 2016). Malfunction of the ependymal cells is a leading cause of the pathophysiological mechanism of hydrocephalus as these cells have a key role in CSF circulation and composition (Ji et al, 2022). In addition, hydrocephalus is linked with disorganization of the subventricular neurogenic niche's cytoarchitecture and neurogenesis defects, as ependymal cells were shown to provide structural and functional support to the adult neural stem cells (Paez-Gonzalez et al, 2011; McAllister et al, 2017; Rodríguez and Guerra, 2017).

Under homeostatic conditions, ependymal cells form one of the largest epithelia of the brain, lying adjacent to the ventricular lumen and bear unique structural characteristics. The cell surface exposed to the ventricular lumen contains a patch of ~50 modified centrioles, called basal bodies, which are tiny microtubule-based organelles responsible for nucleating an equal number of motile cilia. Cilia are anchored on their apical surface, beating in the same direction and in a coordinated fashion, thus contributing to efficient CSF circulation throughout the brain (Mirzadeh et al, 2008, 2010). Constant ciliary beating facilitates secreted molecules dispersion to other regions of the CNS. Such circulating molecules in the CSF influence neurogenesis and neuroblasts' migration to the olfactory bulbs (Lim et al, 2000; Sawamoto et al, 2006).

Ependymal cell differentiation is a multi-step process orchestrated by a tightly regulated transcriptional program which is responsible for activating numerous genes that promote amplification and docking of centrioles and the formation of cilia (Kyrousi et al, 2017; Arbi et al, 2018). We have previously provided evidence that the Geminin family proteins, GEMC1 and MCIDAS, are the earliest regulators for the cell fate commitment to the ependymal lineage (Kyrousi et al, 2015). *GemC1* lies upstream of *McIdas* inducing its transcriptional activation, while both upregulate genes that are essential for centriole amplification and for promoting the downstream transcriptional machinery of multiciliogenesis (Kyrousi et al, 2015; Arbi et al, 2016; Lalioti et al, 2019a). GEMC1 and MCIDAS cooperate with E2F4/5 transcription factors, to upregulate *cMyb* and *Ccno*, which are implicated in centriole amplification (Tan et al, 2013; Wallmeier et al, 2014; Funk et al, 2015; Arbi et al, 2018). In addition, we and others have provided evidence that *GemC1* and *McIdas* drive the transcriptional activation of P73 and FOXJ1 transcription factors, which are implicated in basal bodies docking and cilia motility (Kyrousi et al,

[1]Department of Physiology, School of Medicine, University of Patras, Patras, Greece. [2]Department of General Biology, School of Medicine, University of Patras, Patras, Greece.
✉E-mail: taraviras@med.upatras.gr

2015; Arbi et al, 2016; Nemajerova et al, 2016; Marshall et al, 2016; Lalioti et al, 2019a).

Given the multifaceted roles that ependymal cells have in hydrocephalus pathophysiology, restoration of ependymal cells could significantly contribute to hydrocephalus therapy. With the advancements in direct reprogramming-based strategies for conditions affecting the central nervous system (Bocchi et al, 2022), it is plausible to anticipate that cellular reprogramming directed towards ependymal cells could represent a significant breakthrough in the treatment of hydrocephalus. In the present study we showed that ectopic expression of *GemC1* or *McIdas* promotes the programming of mouse pluripotent embryonic stem cells and reprogramming of cortical astrocytes into ependymal cells, with *McIdas* being more efficient on establishing functional motile cilia in reprogrammed astrocytes. In addition, forced expression of *McIdas* in both a genetic and an acquired hydrocephalus mouse model promoted in vivo reprogramming of cells residing in the ventricular walls of the diseased brain into functional ependymal cells. Importantly, reprogrammed ependymal cells were able to beat their cilia in a coordinated fashion and formed pinwheel structures together with neural stem cells in hydrocephalic mice, revealing their potential to regenerate the architecture of the neurogenic niche, which is disrupted in hydrocephalus. Collectively, our data suggest that GEMC1 and MCIDAS can orchestrate the transcriptional program of multiciliogenesis establishing ependymal cell fate and differentiation, providing proof of principle evidence that restoration of ependymal cells could potentially contribute to hydrocephalus management.

## Results

### GemC1 and McIdas promote programming of mouse embryonic stem cells and reprogramming of cortical astrocytes into ependymal cells

*GemC1* and *McIdas* are the most upstream regulators driving multiciliogenesis in vertebrate organisms (Stubbs et al, 2012; Ma et al, 2014; Boon et al, 2014; Kyrousi et al, 2015; Terré et al, 2016; Arbi et al, 2016). We therefore postulated that *GemC1* and *McIdas* would be capable of inducing stem cell programming towards multiciliated ependymal cells. To examine this hypothesis, mouse embryonic stem cells (mESCs) were transfected with plasmids encoding *McIdas* or *GemC1* in conjunction with GFP (referred to as McIdas and GemC1, respectively), or GFP alone as a control. Transfected mESCs were cultured for 4 days in the absence of Leukemia Inhibitory Factor (LIF) and subjected to immunostaining for FoxJ1, a key transcription factor for the differentiation of ependymal cells, and Pericentrin which marks nascent basal bodies (Fig. EV1A). Our analysis showed that 87% of *McIdas*- and 39% of *GemC1*-overexpressing mESCs were expressing FoxJ1, while only 1% FoxJ1-positive cells were detected in the control (Fig. EV1B). In addition, we identified that 90% of McIdas- and 59% of GemC1-overexpressing mESCs showed accumulation of Pericentrin staining (Fig. EV1C), thus presented multiple nascent basal bodies, corresponding to differentiating ependymal cells. Furthermore, we examined whether McIdas and GemC1-overexpressing mESCs acquire multiple cilia, a unique structural characteristic of the ependymal cells. Double immunofluorescence experiments were

performed 6 days following mESCs transfection using anti-Pericentrin and anti-acetylated α-tubulin antibodies to mark the basal bodies and cilia, respectively (Fig. EV1D). Our analysis revealed that 66% of McIdas- and 21% of GemC1-overexpressing mESCs develop multiple cilia arising from numerous basal bodies. On the contrary, control mESCs develop neither multiple basal bodies nor cilia (Fig. EV1D,E). Our data show that ectopic expression of *McIdas* and *GemC1* programs mESCs into multiciliated ependymal cells.

Next, we sought out to explore the reprogramming potential of *McIdas* and *GemC1* in astrocytes that under physiological conditions do not differentiate into ependymal cells. Astrocytes were isolated from the cortex of neonatal mice and were subsequently infected with lentiviruses expressing GFP-McIdas or GFP-GemC1 (referred to as McIdas and GemC1, respectively), while lentiviruses expressing GFP alone were used as control. Cells were subsequently cultured under differentiating conditions and immunofluorescence experiments were conducted to assess the expression of known ependymal and astrocytic markers (Fig. 1A). Ectopic expression of both *McIdas* and *GemC1* resulted in *P73* expression, a marker of the early steps of ependymogenesis, in 37% and 36% of McIdas- and GemC1-infected cells, respectively, while in the control condition only 4% of the transduced cells was expressing *P73* (Figs. 1B,C and EV2A,B). Notably, concurrent changes in the expression pattern of the astrocytic marker S100β were observed in ~20% of McIdas- and 18% of GemC1-infected astrocytes, as S100β expression was detected around their cell body, which is characteristic for ependymal cells (Mirzadeh et al, 2008) (Figs. 1B–D and EV2A–C). On the contrary, control astrocytes retained a cytoplasmic expression of the S100β protein, a typical feature of astrocytes. Overall, our data suggest that ectopic expression of McIdas and GemC1 can lead to the acquisition of the ependymal cell fate.

Subsequently, we examined whether McIdas- and GemC1-transduced astrocytes successfully differentiate into ependymal cells. We performed immunofluorescence experiments using specific antibodies recognizing FoxJ1 and Pericentrin, to identify astrocytes that acquired the ependymal fate and successfully managed to multiply their basal bodies, respectively. Our analysis showed that 14 days post infection 49% of McIdas- and 19% of GemC1-infected astrocytes were expressing FoxJ1 (Figs. 1E,F and EV2D,E). In addition, 28% of the McIdas- and 12% of GemC1-infected astrocytes possessed multiple basal bodies, indicated by the accumulation of Pericentrin signal (Figs. 1G and EV2F). Moreover, expression of a GFP lentivirus did not result into FoxJ1 expression nor accumulation of Pericentrin.

In addition, we examined the ability of McIdas and GemC1 to repress the astrocytic identity of the transduced astrocytes. Toward this direction, the combined expression of the astrocytic markers GFAP and S100β was assessed through immunofluorescence experiments at 14 days post infection. Our analysis revealed that McIdas ectopic expression resulted to 23% GFAP + /S100β+ double-positive cells compared to 31% in the control condition (Appendix Fig. S1A,B). We did not observe a statistically significant change in the percentage of GFAP+/S100β+ double-positive cells following GemC1 ectopic expression (Appendix Fig. S1C). In addition, we examined the percentage of infected cells that were GFAP- and reveal S100β expression around the cell body, which correspond to ependymal cells. We showed that 28% McIdas- and 22% GemC1-infected astrocytes present this expression pattern (Appendix Fig. S1A,B), which is in accordance with our analysis

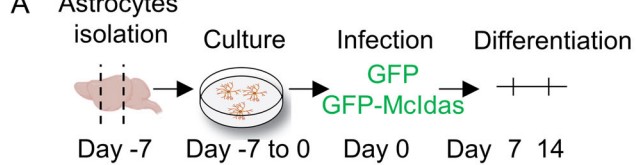

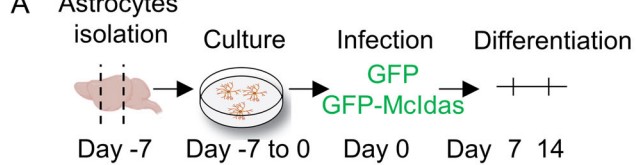

**Figure 1. McIdas forced expression promotes ex vivo reprogramming of murine cortical astrocytes into ependymal cells.**

(A) Schematic representation of experimental procedures for (B–G). Astrocytes were isolated from the cortex of neonatal mice and were cultured under proliferating conditions. Subsequently, cells were transduced with lentiviral vectors encoding GFP or GFP-McIdas (McIdas) and cultured under differentiating conditions for indicated time points. (B) Cortical astrocytes were immunostained against GFP (green) to mark infected cells, P73 (gray) and S100β (red) at day 7 of differentiation. Arrows point to McIdas-infected cells which are P73+ and represent cells committed to the ependymal lineage. (C, D) The percentage of the infected cells which display S100β staining around their cell body, which corresponds to ependymal cells, is depicted in the graph (C). The graph depicts the percentage of the infected cells that co-express the ependymal marker P73 over the total number of the infected cells (D). Data are presented as the median ±interquartile range (IQR) of three independent experiments. Statistical significance was determined using the nonparametric two-tailed Mann–Whitney test (***$P < 0.0001$). (E) Transduced astrocytes were immunostained at differentiation day 14 against GFP (green), FoxJ1 (gray) and pericentrin (PCNT, red). Note the presence of multiple basal bodies, based on PCNT signal accumulation, in McIdas-infected cells which co-express the ependymal marker FoxJ1 (arrows). (F, G) The graphs present the percentage of the infected cells that express FoxJ1 over the total number of the infected cells (F) and the percentage of FoxJ1-positive infected cells which also display accumulation of PCNT signal (G). Data are presented as the median ±IQR of three independent experiments. Statistical significance was determined using the nonparametric two-tailed Mann–Whitney test (***$P < 0.0001$). DNA was stained with DAPI (blue). Scale bars, 10 μm. dd differentiation day. Source data are available online for this figure.

performed in earlier stages of reprogramming (Fig. 1D). Our results indicate that a significant percentage of the infected astrocytes are reprogrammed to the ependymal lineage and possess unique ependymal morphological characteristics. Moreover, we showed that McIdas is more potent in the downregulation of the astrocytic identity of the transduced cells.

GemC1 is positioned upstream of McIdas in the molecular pathway governing ependymogenesis, as it regulates the transcriptional activation of McIdas and controls the initial fate commitment towards the ependymal lineage (Kyrousi et al, 2015, 2017). Our results suggest that GemC1 was less effective than McIdas in reprogramming of astrocytes. We have therefore examined whether

ectopic expression of GemC1 in astrocytes would not induce McIdas' expression at sufficient levels. To test this hypothesis, we conducted immunofluorescence experiments using a McIdas-specific antibody and quantified McIdas fluorescence in McIdas- and GemC1-infected astrocytes 7 days post infection (Appendix Fig. S2A). Our results revealed that McIdas' ectopic expression led to higher expression levels compared to GemC1 (Appendix Fig. S2B), providing a potential mechanism for their differential ability to induce reprogramming.

Given the greater capacity of *McIdas* to induce the ependymal differentiation in astrocytes compared to *GemC1*, we hypothesized that *McIdas* would reprogram cortical astrocytes into fully mature ependymal cells carrying multiple basal bodies nucleating motile cilia. To address this point, we initially performed immunofluorescence experiments in transduced astrocytes 22 days post infection using antibodies against Centriolin (also known as Cep110), a known centrosomal marker, and Meig1 which was recently characterized as ependymal cell marker (MacDonald et al, 2021). We detected accumulation of both Centriolin (Appendix Fig. S3A) and Meig1 (Appendix Fig. S3B) signal in McIdas-infected astrocytes, verifying the generation of multiple basal bodies upon McIdas' ectopic expression. Next, we performed immunofluorescence experiments making use of antibodies against acetylated α-tubulin, which stains ciliary axonemes, and the astrocytic marker glial fibrillary acidic protein (GFAP). The ectopic expression of *McIdas* resulted in the downregulation of GFAP expression and the generation of multiple cilia on the surface of the reprogrammed astrocytes, revealing the loss of the astrocytic identity and the establishment of the ependymal differentiation (Fig. 2A). Our analysis showed that 18% of McIdas-infected astrocytes were reprogrammed into multiciliated ependymal cells based on acetylated α-tubulin staining (Fig. 2C), which was consistent with the percentage of McIdas-infected cells that downregulate the expression of the astrocytic marker GFAP (Fig. 2B). In addition, live imaging microscopy experiments were performed in transduced astrocytes 22 days post infection to address the ability of the reprogrammed astrocytes multicilia to beat (Movies EV1–4). Using high-speed video microscopy, we acquired fast video recordings from GFP (Movies EV1 and 2) and McIdas-infected astrocytes (Movies EV3 and 4), which revealed the presence of ciliary motility in McIdas-infected cells as opposed to the absence of motility in GFP-infected cells. Importantly, our analysis showed that McIdas expression established cilia movement that was sufficient to propel fluorescent particles added to the imaging medium, highlighting the functionality of the ependymal cells and their capability to create a fluid flow. On the contrary, astrocytes infected with a control virus were unable to propel the fluorescent particles, as were only observed to have a rather stable forward and backward motion (Fig. 2D).

Our findings show that GemC1 and McIdas promote the early steps of the multiciliogenesis program in both embryonic stem cells and astrocytes. Importantly, McIdas has a greater capacity for eliciting direct reprogramming into functional ependymal cells.

## McIdas induces ependymal cells regeneration in a mouse model of intracranial hemorrhage hydrocephalus

In both human and murine hydrocephalus, the disrupted ependymal cell population is replaced by astrocytes. However, this astrocytic scarring along the denuded ventricular wall fails to restore the ependymal functions (Sival et al, 2011; Roales-Buján

et al, 2012; Guerra et al, 2015; McAllister et al, 2017). Based on our finding that *McIdas* can successfully reprogram cortical astrocytes into functional ependymal cells, we examined whether *McIdas* could induce direct reprogramming of periventricular cells in hydrocephalic models. We used an established mouse model of intracranial hemorrhage hydrocephalus induced by the elevated concentration of Lysophosphatidic Acid (LPA) in the brain (Yung et al, 2015; Lummis et al, 2019), which closely mirrors the most common type of human hydrocephalus.

Based on previously established protocols (Yung et al, 2011; Lummis et al, 2019), postnatal day 5 (P5) mice received intracranial injections of LPA and mouse brains were examined 2 days later to evaluate hydrocephalus occurrence. Although no obvious differences were observed in the size of LPA-treated brains macroscopically (Fig. EV3A), coronal brain sections revealed ventricular dilation in LPA-injected animals compared to non-injected ones (Fig. EV3B), confirming the development of hydrocephalus. In addition, acetylated α-tubulin immunofluorescence revealed the ciliary disruption in ependymal cells alongside the ventricular walls upon LPA administration (Fig. EV3C), in line with previous studies (Lummis et al, 2019). On the contrary, ependymal cells with multiple cilia covered the entire ventricular walls in control animals (Fig. EV3C). Given that the ventricular walls of the LPA-injected hydrocephalic mice were denuded from ependymal cells, we wished to determine their cellular composition. We performed immunofluorescence with an antibody against the astrocytic marker glial fibrillary acidic protein (GFAP). Our analysis showed that GFAP-positive cells with long processes, which likely corresponded to reactive astrocytes, covered the walls of the lateral ventricles in regions where ependymal cells localize under physiological conditions (Fig. EV3C), in line with previous findings (Roales-Buján et al, 2012).

To assess whether McIdas can induce direct reprogramming into ependyma in LPA-induced hydrocephalus, GFP-McIdas (McIdas) or GFP-expressing plasmids were electroporated onto P7 brains isolated from LPA-treated mice by targeting the dorsolateral wall of one lateral ventricle to introduce plasmids in cells located at periventricular regions. Subsequently, thick coronal brain sections at the level of the lateral ventricles were cultured, and immunofluorescence was performed using established ependymal markers (Fig. 3A). Five days post electroporation, 77% of the McIdas-electroporated cells expressed *P73*, as opposed to 13% of the GFP-electroporated cells, suggesting that astrocytes that replaced damaged ependymal cells in LPA-treated hydrocephalic mouse models acquired the ependymal fate (Fig. 3B).

As ependymal cells normally require approximately 2 weeks until they acquire their mature characteristics in vivo, we followed the differentiation of electroporated cells at later time points and examined whether they differentiated into fully mature multiciliated cells. Towards this direction, we performed immunolabeling experiments making use of antibodies recognizing Pericentrin to mark basal bodies and acetylated α-tubulin to detect cilia (Fig. 3C). Nine days post electroporation 67% of McIdas-electroporated cells carried multiple basal bodies (Fig. 3D), while 16% of McIdas-electroporated cells carried multiple cilia as revealed by acetylated α-tubulin immunostaining (Fig. 3E). Basal bodies amplification or multiple cilia formation was not detected in any of the control cells, electroporated with a GFP-expressing plasmid.

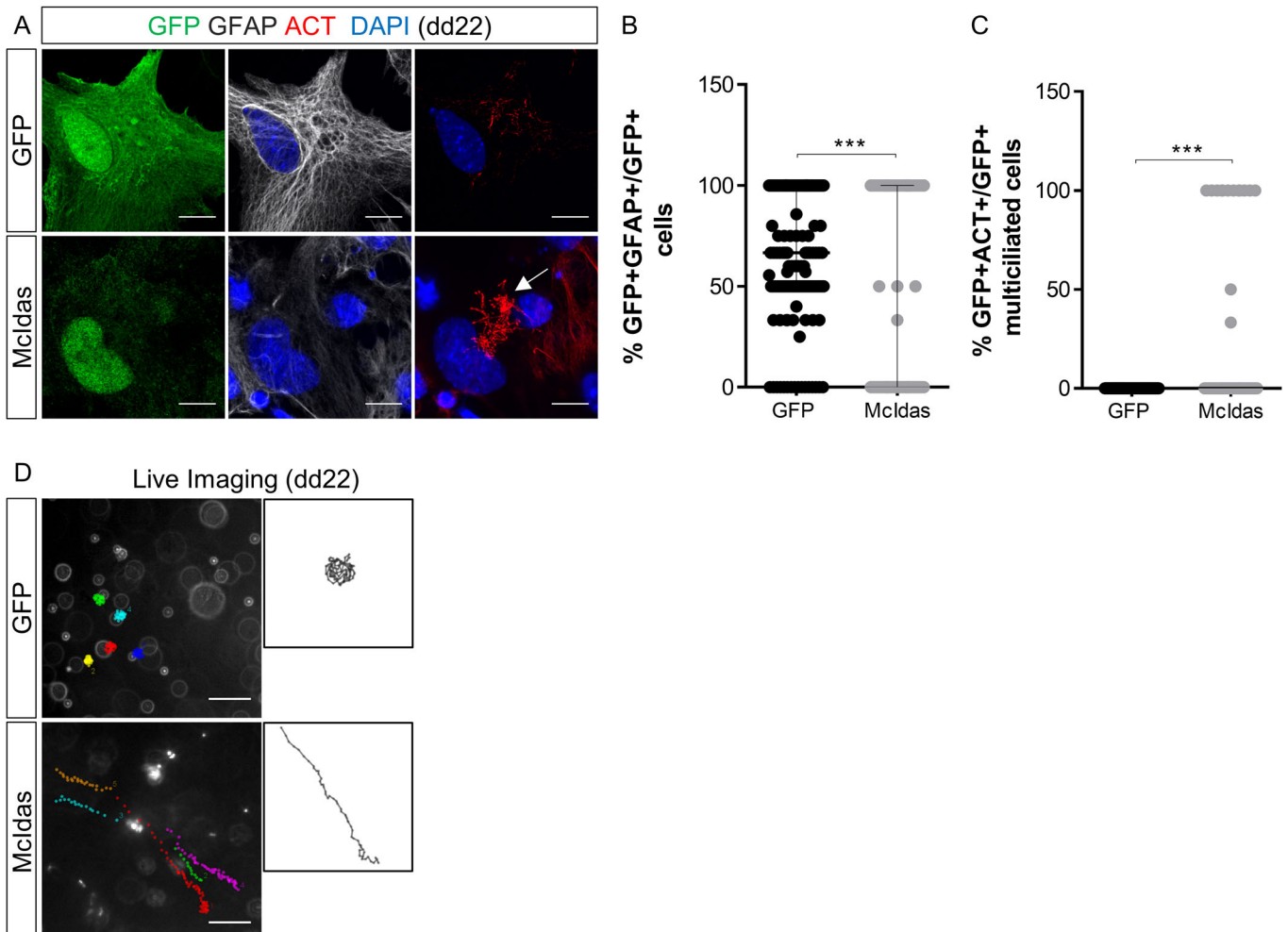

**Figure 2. McIdas reprograms cortical astrocytes into functional ependymal cells.**

(A) Immunofluorescence of transduced cortical astrocytes at differentiation day 22 with antibodies against GFP (green), Glial Fibrillary Acidic Protein (GFAP, gray) which labels astrocytes and acetylated α-tubulin (ACT, red), a marker of ciliary axonemes. McIdas expression triggers the formation of multiple cilia, tiny hair-like structures which appear on the surface of the reprogrammed astrocytes (arrow). Note the downregulation of the GFAP signal in the reprogrammed multiciliated cells. DNA was stained with DAPI (blue). Scale bars, 10 μm. (B, C) The graph shows the percentage of infected cells that express GFAP and correspond to an astrocytic population. (B) The percentage of the infected cells which possess multiple cilia based on ACT staining is depicted in the graph. (C) Data are presented as the median ±IQR of two independent experiments. Statistical significance was determined using the nonparametric two-tailed Mann–Whitney test (***P < 0.0001). (D) Live imaging microscopy experiments were performed on cortical astrocytes transduced either with GFP or McIdas lentiviruses. The positions of fluorescent beads included in the live imaging medium, were marked with colored dots on selected video frames. Track of one representative bead is shown in the right panel. Two independent experiments were performed. Scale bar, 10 μm. dd differentiation day. Source data are available online for this figure.

Collectively, our data show that *McIdas* forced expression in brain periventricular cells has the potential to regenerate the ependymal cells in a mouse model of intracranial hemorrhage hydrocephalus.

## McIdas promotes in vivo reprogramming towards functional ependymal cells in a model of congenital hydrocephalus

We have recently described that genetic mutations in the human and mouse *GemC1* gene lead to hydrocephalus (Lalioti et al, 2019b). Moreover, we have shown that radial glial cells lacking functional *GemC1* fail to commit to the ependymal lineage and therefore do not develop ependymal cells during embryogenesis.

We took advantage of the *GemC1*-knockout genetic mouse model of hydrocephalus we have generated to investigate whether reprogramming towards the ependymal cells could be achieved in a model of congenital hydrocephalus.

We initially examined whether *McIdas* could program radial glial cells derived from *GemC1*-knockout hydrocephalic mice into ependymal cells ex vivo. Postnatal radial glial cells isolated from *GemC1*-knockout mice were infected with lentiviruses expressing either a GFP-McIdas (McIdas) fusion protein or GFP as a control. Infected cells were analyzed at different time points for the expression of known ependymal markers by immunofluorescence experiments, using specific antibodies recognizing FOXJ1, Pericentrin and acetylated α-tubulin proteins (Fig. EV4A). At 5 days post infection, 86% of the McIdas-infected cells expressed *FoxJ1* and showed accumulation of

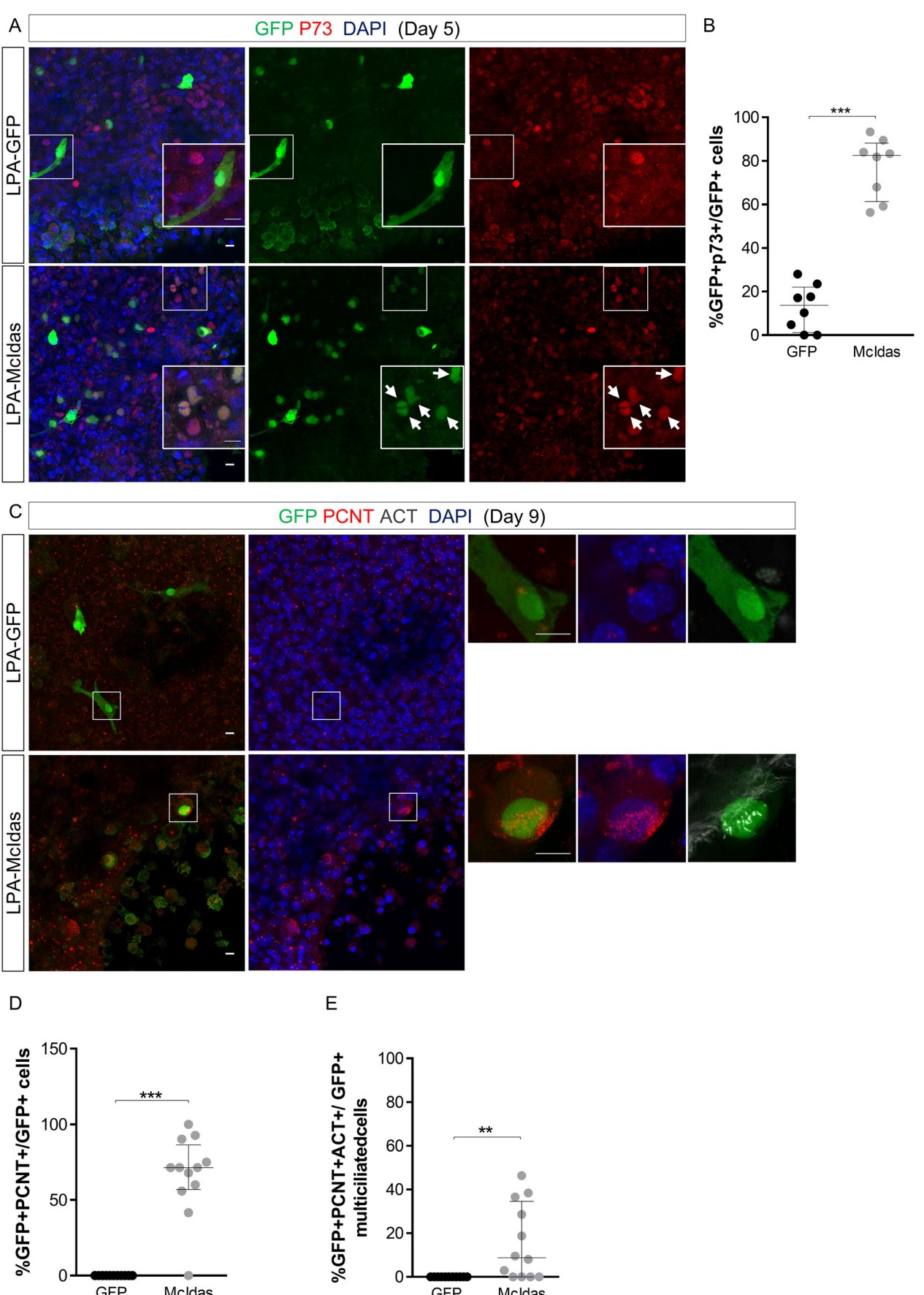

**Figure 3.  McIdas induces ependymal cell regeneration in a mouse model of intracranial hemorrhage hydrocephalus.**

(A) Mice received unilateral intracranial injection of Lysophosphatidic acid (LPA) to induce hydrocephalus. Coronal brain slices derived from brains electroporated either with a GFP or GFP-McIdas (McIdas) plasmid were cultured for 5 days. Immunofluorescence experiments were performed with antibodies against GFP (green) and the ependymal marker P73 (red). Inserts show higher magnification images. Arrows indicate GFP + P73+ cells. (B) The graph depicts the percentage of the electroporated cells that co-express the ependymal marker P73, over the total number of the electroporated cells, at day 5 of culture. Data are presented as the median ± IQR of three independent experiments. Statistical significance was determined using the nonparametric two-tailed Mann–Whitney test (***$P = 0.0002$). (C) Coronal brain slices were cultured for 9 days and immunostained against GFP (green), pericentrin (PCNT, red) and acetylated α-tubulin (ACT, gray) to detect mature multiciliated cells. Note the presence of multiple basal bodies and cilia in McIdas-electroporated cells. (D, E) The percentage of the electroporated cells that carry multiple basal bodies, based on PCNT accumulation signal, upon 9 days of culture, is presented in the graph (D) (***$P < 0.0001$). The graph shows the percentage of multiciliated cells based on PCNT and ACT staining (E) upon 9 days of culture ($P = 0.0013$). Data are presented as the median ±IQR of three independent experiments. Statistical significance was determined using the nonparametric two-tailed Mann–Whitney test. DNA was stained with DAPI (blue). Scale bars, 10 μm. Source data are available online for this figure.

multiple basal bodies, based on Pericentrin immunostaining (Fig. EV4B,C). Moreover, our analysis revealed that 97% of the McIdas transduced cells were multiciliated 15 days following infection, based on acetylated α-tubulin and Pericentrin staining (Fig. EV3D,E). By contrast, in GFP transduced cells neither *FoxJ1* expression nor multiple basal bodies and cilia were observed. In addition, similar experiments were performed by ectopically expressing *P73* in postnatal radial glial cells isolated from *GemC1*-knockout mice. Ectopic expression of *P73* was not sufficient to induce *FoxJ1* expression nor multiple basal bodies or multiple cilia formation in *GemC1*-knockout radial glial cells (Fig. EV4A–E). These results show that cells derived from a genetic model of hydrocephalus can be programmed ex vivo into multiciliated ependymal cells.

In order to examine whether *McIdas* promotes cellular reprogramming towards the ependymal cell lineage in vivo, we have used our genetic mouse model of hydrocephalus which was previously established (Lalioti et al, 2019b). Plasmid vectors encoding *McIdas* in conjunction with GFP or GFP alone were electroporated into one lateral ventricle of newborn *GemC1*-knockout mouse brains targeting periventricular cells which could be either neural progenitor cells or reactive astrocytes that have replaced the lost ependymal cell population (Roales-Buján et al, 2012; Gonzalez-Cano et al, 2016; Abdi et al, 2018; Lalioti et al, 2019b). Reprogramming towards the ependymal cells was evaluated through immunofluorescence, using molecular markers that determine different stages of ependymogenesis. Interestingly, 4 days post electroporation 61% and 58% of McIdas-electroporated cells were immunopositive for the ependymal markers *P73* (Fig. 4A,B) and FoxJ1, respectively (Fig. EV5A,B). The expression of *P73* and *FoxJ1* was not detected in control animals, in agreement to what was previously described (Lalioti et al, 2019b), as *GemC1*-knockout mice lack ependymal cells. We also showed that *McIdas* further promoted the differentiation of the electroporated ependymal cells as basal bodies amplification was observed, based on Pericentrin immunofluorescence (Fig. 4C). More specifically, upon *McIdas* overexpression 44% of the electroporated cells acquired multiple basal bodies (Fig. 4D). In addition, immunofluorescence against acetylated α-tubulin revealed that *McIdas* overexpression resulted in the formation of cilia in a significant number of cells, corresponding to 7% of McIdas-electroporated cells (Fig. 4E). Noticeably, ciliated cells were located close to the ventricles, resembling normal mature multiciliated ependymal cells, which form a single-layered ciliated epithelium adjacent to the brain ventricle.

In order to examine whether the cilia that were generated upon *McIdas* ectopic expression were motile, thick coronal brain sections from *GemC1*-knockout mice covering the lateral brain ventricles were obtained and used in live imaging microscopy experiments

(Fig. 5A). Electroporated cells were identified under a FITC filter and their cilia beating capacity was examined in fast video records. A proportion of cells with multiple beating cilia that were GFP-negative in sections derived from McIdas-electroporated brains were observed. We have previously shown that GemC1-knockout mice are unable to generate ependymal cells (Lalioti et al, 2019b), therefore cells carrying multiple beating cilia that do not appear GFP-positive were possibly observed due to the loss of the transgene because of its transient expression and were thus undetectable under the FITC filter. In addition to this, we cannot exclude the possibility of dilution of the GFP signal due to continued proliferation. In agreement with what was previously described cilia were not detected in brain sections following GFP overexpression. However, McIdas-electroporated cells exhibited functional beating cilia (Fig. 5B; Movies EV5 and 6). Kymographs were generated (Fig. 5C) following previously described methods by Francis et al (Francis and Lo, 2013) and ciliary beating frequency of McIdas-induced ependymal cells was estimated to have a mean value of 10 Hz (Fig. 5D).

Our findings show that McIdas induces ependymal cell restoration in the diseased hydrocephalic brain.

## McIdas expression contributes to the restoration of the subventricular zone niche cytoarchitecture in hydrocephalic mice

The adult subventricular zone niche has a well-defined architecture formed by pinwheel-like structures. These structures are assembled by the large apical domain of ependymal cells which surround the smaller apical domain of adult neural stem cells (aNSCs) and constitute a unique microenvironment for coordinating adult neurogenesis (Mirzadeh et al, 2008; Paez-Gonzalez et al, 2011; Kokovay et al, 2012). To assess whether McIdas is able to restore the subventricular zone niche's cytoarchitecture in hydrocephalic mice, whole-mount immunofluorescence was performed on ventricular walls of the lateral ventricles of GemC1-knockout hydrocephalic mice at P7-P9, previously electroporated with an *McIdas*-expressing plasmid or GFP alone. Specific antibodies against VCAM1 and GFAP, both markers of aNSCs, and β-catenin to delineate cell boundaries were used. Our analysis showed that McIdas-electroporated cells had a larger apical surface compared to control GFP-electroporated cells, which is characteristic for multiciliated ependymal cells. In addition, McIdas-electroporated cells were surrounding VCAM1- (Fig. 6A) and GFAP-immunopositive cells (Fig. 6B) reminiscent of pinwheel structures. On the contrary, control GFP-electroporated cells had a

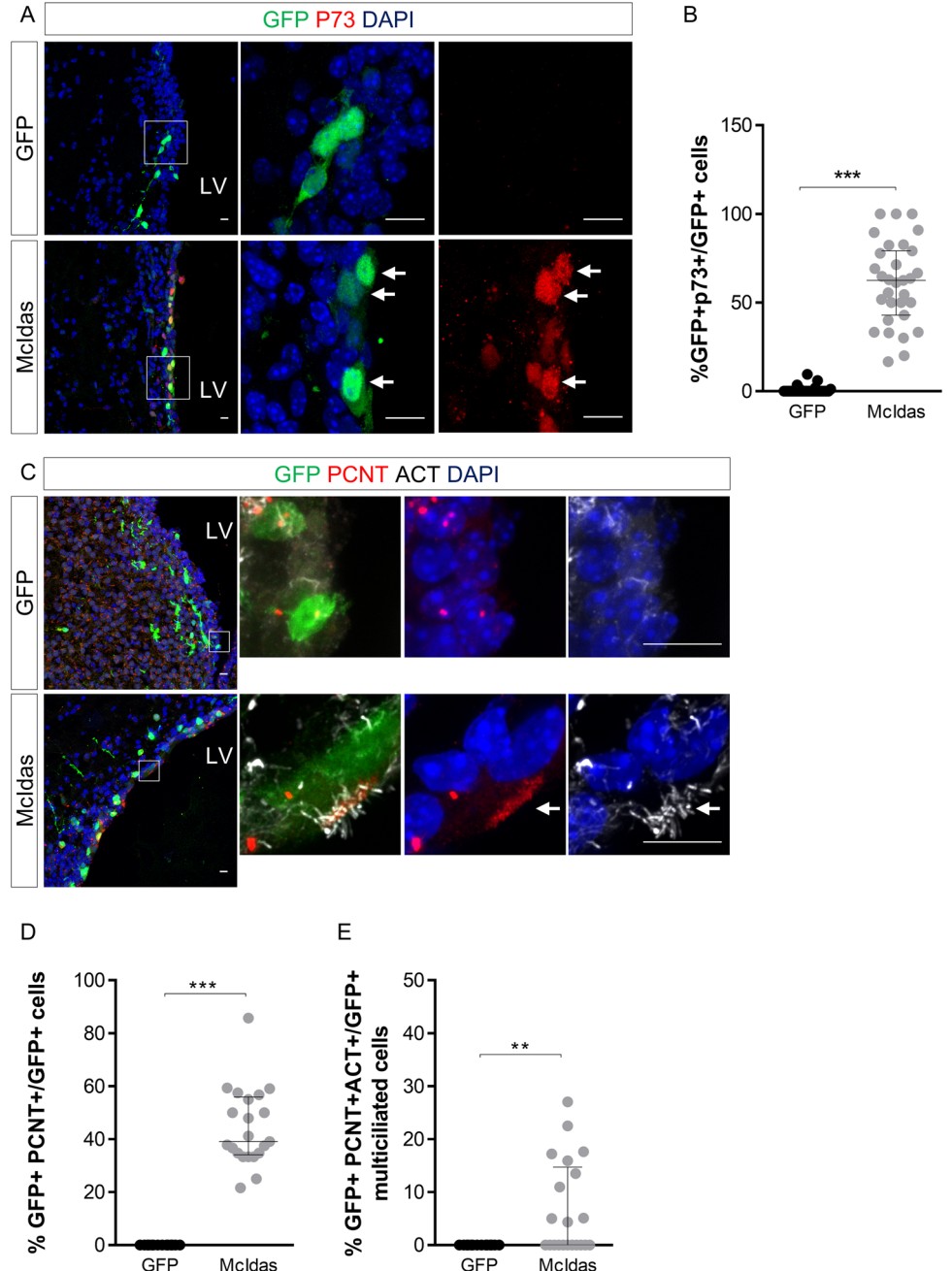

**Figure 4. McIdas ectopic expression promotes in vivo reprogramming into ependymal cells in a genetic mouse model of hydrocephalus.**

(**A**) Neonatal subventricular zone electroporation was conducted at GemC1-knockout mice at postnatal day 1 (P1) with plasmids encoding GFP, or GFP-McIdas (McIdas). Mice were sacrificed 4 days post electroporation and coronal brain sections were immunostained with antibodies against GFP (green) and P73 (red). Arrows point to McIdas-electroporated cells co-expressing P73, which represent cells committed to the ependymal lineage. Higher magnification of the boxed region is shown in the right panel. (**B**) The graph shows the percentage of the electroporated cells expressing P73. Data are presented as the median ±IQR of three independent experiments. Statistical significance was determined using the nonparametric two-tailed Mann–Whitney test (***P < 0.0001). (**C**) Coronal brain sections from GemC1-knockout electroporated mice were stained with specific antibodies against GFP (green), Pericentrin (PCNT, red) and acetylated α-tubulin (ACT gray). Multiple basal bodies and cilia on the surface of McIdas-electroporated cells are indicated by an arrow. Higher magnification of the boxed region is shown in the right panel. (**D, E**) Quantification of the percentage of the electroporated cells with multiple basal bodies, based on PCNT accumulation (**D**) (***P < 0.0001), and the percentage of multiciliated cells based on PCNT and ACT staining over the total number of the electroporated cells (**E**) (P = 0.0041). Data are presented as the median ±IQR of two independent experiments. Statistical significance was determined using the nonparametric two-tailed Mann–Whitney test. DNA was stained with DAPI (blue). Scale bars, 10 μm. LV lateral ventricle. Source data are available online for this figure.

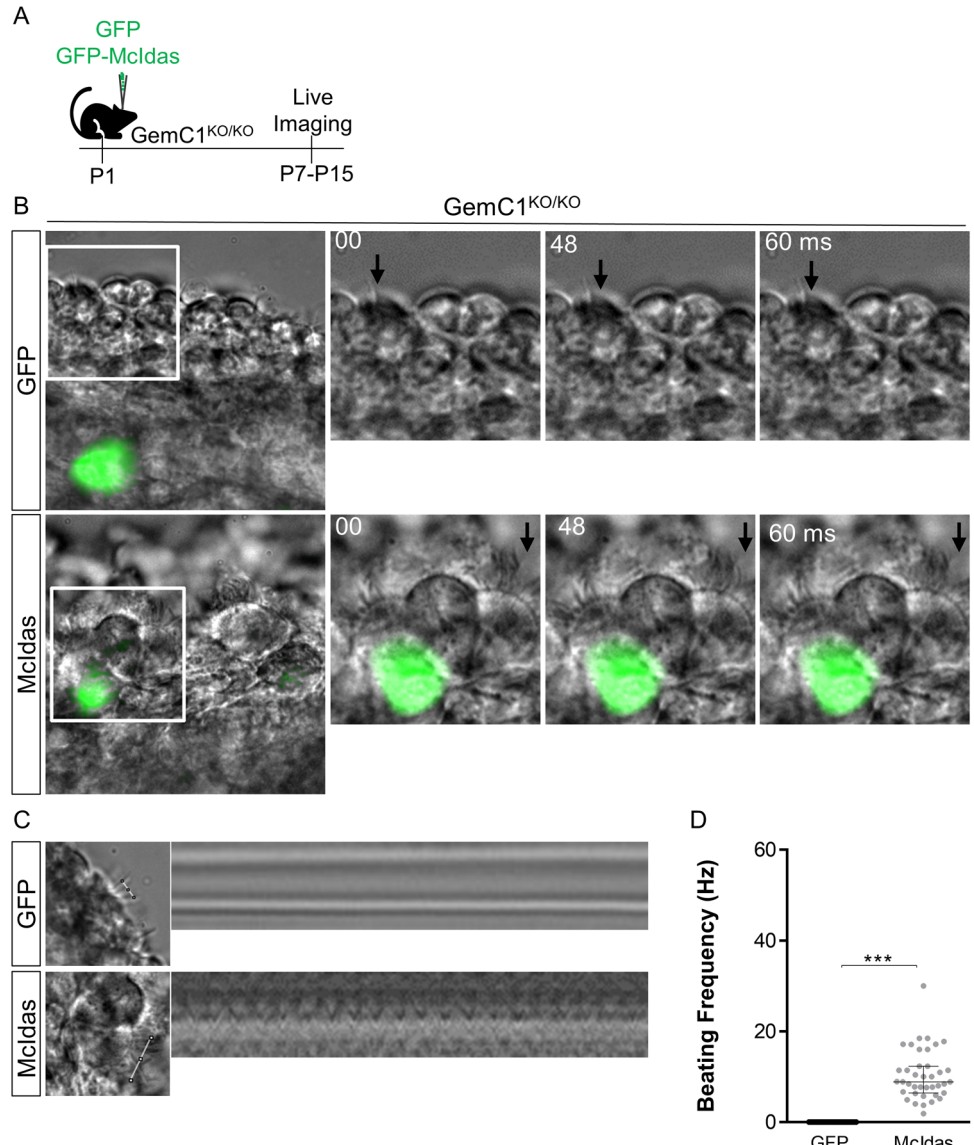

**Figure 5. McIdas establishes cilia motility in reprogrammed cells in a genetic mouse model of hydrocephalus.**

(A) Schematic representation of experimental procedures for (B–D). GFP or GFP-McIdas (McIdas) expressing plasmids were electroporated at the lateral ventricle of postnatal day 1 (P1) GemC1-knockout mouse brains. Coronal brain sections, that contained the lateral ventricles, were obtained from the electroporated mice at ages P7–P15 and were used in live imaging experiments. (B) Cilia still images from GFP or McIdas-electroporated brains at indicated time points in milliseconds. FITC fluorescent cells are presented together with brightfield images. Arrows indicate the positions of cilia at indicated time points. Cilia motility was observed upon McIdas ectopic expression as evidenced by cilia changing positions at indicated time points. Higher magnification of the boxed regions is shown in the right panels. (C) Representative kymograms depicting cilia motility upon GFP or McIdas ectopic expression. Left panel images show the lines used to obtain the kymograms. These images are extracted from the image shown in (B), which was utilized to generate the kymograph. (D) The graph shows cilia beat frequency of electroporated cells quantified from high-speed brightfield cilia motility movies. Data are presented as the median ±IQR (n = 2 for GFP (control) and n = 4 for McIdas). Statistical significance was determined using the nonparametric two-tailed Mann–Whitney test (***P < 0.0001). Source data are available online for this figure.

small apical surface and pinwheel structures were not detectable as it was previously described (Lalioti et al, 2019b). We also quantified the number of electroporated cells that were detected in pinwheel structures, revealing that 18% of McIdas-expressing cells were able to form pinwheel structures as opposed to GFP-electroporated cells that were not detected in pinwheels (Fig. 6C). Our data suggest that McIdas expression is sufficient to instruct the regeneration of ependymal cells in hydrocephalic mice which have the potential to re-build the lost architecture of the subventricular zone niche.

## Discussion

During mammalian brain development neural progenitor cells transition from an uncommitted to a restricted state, through a process that is influenced by lineage-specific gene expression programs, resulting to terminally differentiated cells. Reprogramming allows the conversion of differentiated cells from one lineage to another type of differentiated cell. Efficient reprogramming is highly dependent on the selection of pioneer factors related to the

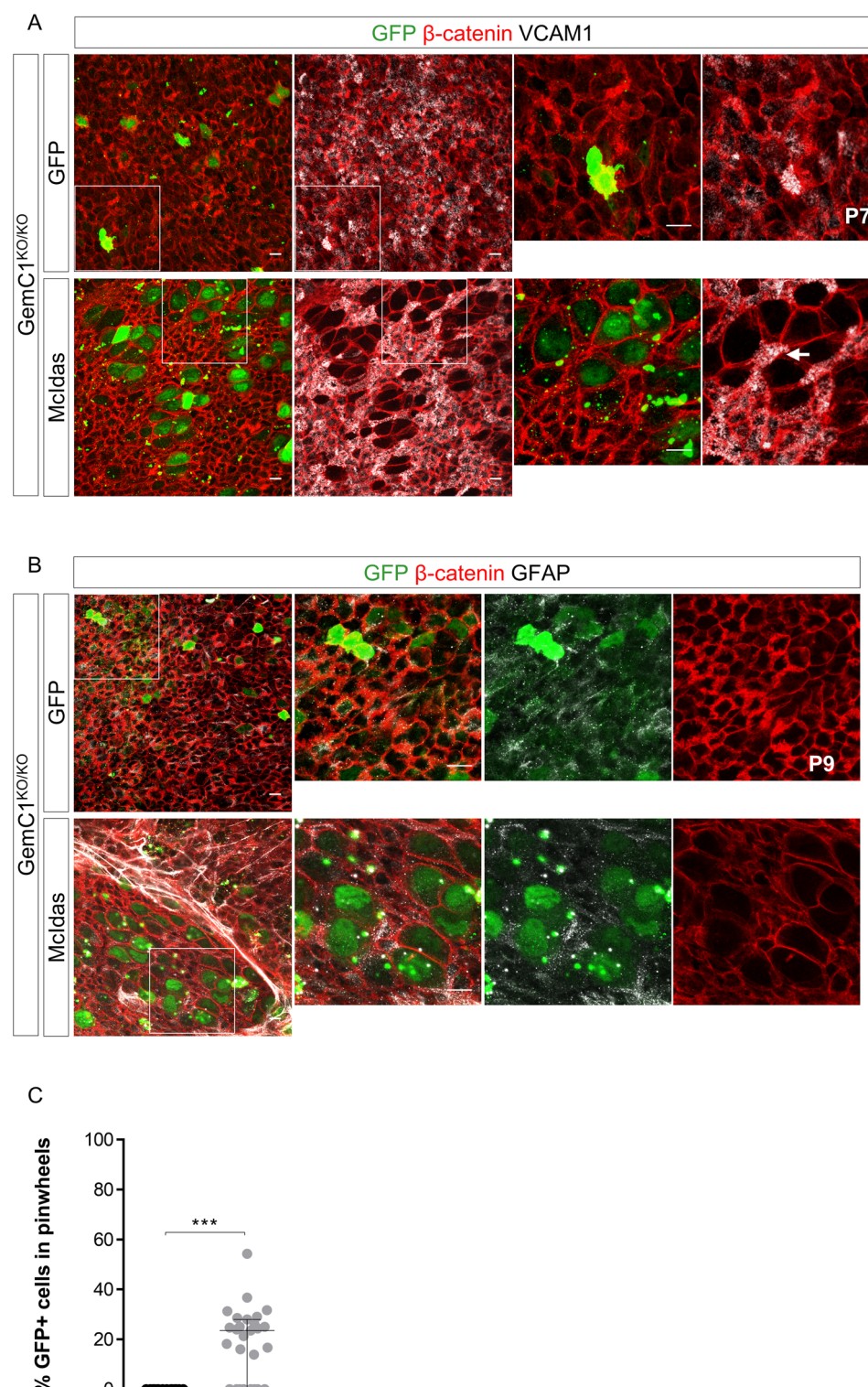

desired cell type. Here, we demonstrate that GEMC1 and MCIDAS, members of the geminin family, constitute potent reprogramming factors toward the ependymal fate both ex vivo and in vivo.

*GemC1* and *McIdas* drive the cell fate commitment of a subpopulation of brain radial glial cells towards multiciliated ependymal cells by coordinating gene expression of essential factors for ependymal cell differentiation (Kyrousi et al, 2015). In addition, we have recently provided evidence that *GemC1* expression is necessary and sufficient to establish the appropriate chromatin organization at multiple loci for ependymal cells differentiation

**Figure 6. Ectopic expression of McIdas restores the subventricular zone niche cytoarchitecture.**

(A, B) GemC1-knockout mouse brains were electroporated with plasmids encoding GFP or McIdas-GFP (McIdas) at postnatal day 1 (P1). Mice were sacrificed at ages P7-P9 and lateral ventricular wall whole-mount immunofluorescence was conducted with antibodies against GFP (green), β-catenin (red), and Vascular Cell adhesion Molecule 1 (VCAM1, gray) (A) or Glial Fibrillary Acidic Protein (GFAP, gray) (B). Formation of pinwheel structures by McIdas-reprogrammed ependymal cells surrounding VCAM1+ neural stem cells (arrow, A) or GFAP-expressing neural stem cells (B) was observed. Note the enlarged apical domain of McIdas-reprogrammed ependymal cells. Higher magnification of the boxed region is shown in the right panel. (C) The graph shows the percentage of electroporated cells, shown in A, identified in pinwheel structures. Data are presented as the median ±IQR ($n = 3$ for GFP (control) and $n = 4$ for McIdas). Statistical significance was determined using the nonparametric two-tailed Mann–Whitney test (***$P < 0.0001$.). Scale bars, 10 μm. Source data are available online for this figure.

(Lalioti et al, 2019b). In light of this evidence, we hypothesized that *GemC1* and *McIdas* could promote direct reprogramming towards the ependymal lineage in different cellular populations under physiology and disease. We initially demonstrated that both GemC1 and McIdas overexpression were able to trigger the differentiation of mESCs into ependymal cells, as evidenced by the upregulation of key factors of ependymogenesis and the generation of multiple cilia in the GemC1- and McIdas-overexpressing cells. We also showed that *GemC1* and *McIdas* overexpression induce the expression of *P73* and *FoxJ1*, two well-established transcription factors of the molecular pathway of multiciliogenesis, in cultured cortical astrocytes. Importantly, we demonstrated that *GemC1* and *McIdas* activate the ependymal fate and induce the formation of multiple basal bodies and cilia in the reprogrammed cells. The ectopic expression of GemC1 resulted in a lower percentage of cells with multiple basal bodies, a unique structural feature of ependymal cells, revealing a differential reprogramming potential of the two factors. We showed that GemC1 does not induce sufficient levels of McIdas expression, that could contribute to the reduced ability of GemC1 to execute as efficiently the ependymal differentiation program. Moreover, using video microscopy experiments we showed that *McIdas* ectopic expression was sufficient to reprogram cortical astrocytes into fully functional multiciliated ependymal cells, as reprogrammed cells possessed multiple beating cilia that were able to beat in a coordinated way and generate fluid flow. In addition, McIdas overexpression was sufficient to induce the ependymal fate in radial glial cells derived from the brain of hydrocephalic mice in great contrast to *P73*, further highlighting McIdas' ability to induce the ependymal cell fate. The stem cell properties of the mESCs and radial glial cells used in our study are likely to facilitate the adoption of the ependymal fate. Since this aligns with their differentiation spectrum their conversion to ependyma could be considered as differentiation into the ependymal cell lineage.

It has been shown that in human and experimental hydrocephalus, disrupted ependyma is replaced by astroglial cells (Sival et al, 2011; Roales-Buján et al, 2012; Guerra et al, 2015; McAllister et al, 2017). This astrocytic scarring along the denuded ventricular wall contributes to the pathophysiology of the disease as astrocytes do not restore ependymal functions; as they do not display cilia, produce appropriate trophic signals, or facilitate the production of neurons (Sival et al, 2011; Roales-Buján et al, 2012). Therefore, our strategy revealing the successful conversion of isolated cortical astrocytes into ependymal cells is of particular importance for the treatment of hydrocephalus.

To investigate whether *McIdas* would also reprogram cells in vivo, we used a mouse model of congenital hydrocephalus in which *GemC1* has been inactivated, and a model of acquired hydrocephalus, triggered by intracranial hemorrhage. It was previously shown that *GemC1* deficiency in mice results in the loss of the ependymal cells due to failure on activating the molecular program for the cell fate commitment towards the ependymal lineage (Terré et al, 2016; Lalioti et al, 2019b). Moreover, *GemC1* deficiency has been recently associated with congenital hydrocephalus in both humans and mice (Lalioti et al, 2019b). In this study, we showed that *McIdas* forced expression induces the upregulation of *P73* and *FoxJ1* in brain periventricular cells, which are likely to be neural progenitor cells or/and reactive astrocytes that have replaced the ependymal cell population in the hydrocephalic brain. The activation of the *P73* and *FoxJ1* genes uncovered the ability of *McIdas* to promote the initial steps of the ependymal cell's differentiation program in the hydrocephalic brain. We also showed that McIdas-overexpressing cells generated multiple basal bodies and cilia, both of which are unique features of mature multiciliated ependymal cells, thus revealing the successful reprogramming towards ependyma. Importantly, the multiple cilia of the reprogrammed cells presented functional activity, as we were able to detect beating cilia upon in vivo overexpression of *McIdas* in the hydrocephalic mice. Using the LPA-induced intracranial hydrocephalus mouse model, we demonstrated McIdas ability to convert periventricular cells into ependymal cells. Previous study has shown that LPA causes substantial depletion of the ependymal monolayer, loss of ependymal cilia and subsequent ependymal apoptotic cell death (Lummis et al, 2019). In line with these findings, we did not observe multiciliated ependymal cells in periventricular regions after LPA treatment. Instead, we observed GFAP-positive cells, likely reactive astrocytes, covering the lateral ventricle walls where ependymal cells typically reside under normal conditions, as it has been previously shown in hydrocephalic mice (Páez et al, 2007; Roales-Buján et al, 2012). Therefore, it is highly likely that McIdas reprograms astrocytes into ependymal cells in LPA-treated mice. However, we cannot rule out the possibility that McIdas might also enhance the survival of a subset of ependymal cells resistant to LPA-induced apoptosis.

Collectively, in our experiments, we employed diverse cellular systems and technical approaches, including plasmid and lentiviral vectors, to induce the overexpression of McIdas and GemC1 suggesting that both GemC1 and McIdas can induce the programming or reprogramming of different cellular populations irrespectively of the cell system and vector used. While our approach in the genetic mouse model of hydrocephalus and the LPA-induced intracranial hydrocephalus likely targets periventricular cells, which are neural progenitor cells or/and reactive astrocytes, we cannot rule out the possibility that other cellular populations might also be targeted. Recent evidence from

reprogramming experiments in the mouse retina and the brain indicates that the insert sequences of overexpression constructs can alter the specificity of the construct's promoter, introducing a bias toward the initial cell population being targeted (Wang et al, 2021; Le et al, 2022). In addition, future genetic lineage analysis using transgenic mice to label neural progenitors and/or astrocytes would be needed to clarify which cell population is differentiated into ependymal cells in our approach.

One of the major clinical features in human patients with hydrocephalus is the ventricular zone disruption phenomenon, characterized by the loss of neural stem cells and ependyma (McAllister et al, 2017). Of note, sustained neuron generation depends on the maintenance of the proper cytoarchitecture of the Ventricular-Subventricular Zone (V-SVZ) niche of the brain, which requires preserving the structure of the ependymal cell monolayer intact (Paez-Gonzalez et al, 2011). In this study, we targeted V-SVZ-resident cells in the hydrocephalic brain in which *McIdas* overexpression was induced and observed that McIdas-overexpressing cells populated areas of the ventricular zone niche next to the lateral ventricles, where ependymal cells normally reside under homeostasis. We have used SVZ electroporation to target the periventricular cells of one lateral ventricle of the hydrocephalic brain of GemC1-knockout mice, which results to a limited number of electroporated cells. Therefore, following this experimental approach, the small number of ependymal cells which is generated is not possible to contribute to a significant reduction in the volume of the lateral ventricles as a readout of resolved or reduced hydrocephalus. It has been described that disruption of the ventricular/subventricular zone (V/SVZ) is a critical and consistent factor in the pathophysiology of hydrocephalus (McAllister et al, 2017; Jiménez et al, 2014). In our study, we showed that McIdas ectopic expression contributes to the restoration of the lost cytoarchitecture of the niche in hydrocephalic mice, as we identified reprogrammed ependymal cells forming pinwheel-like structures with resident neural stem cells in the targeted ventricular wall. Our findings suggest that McIdas forced expression in human patients with hydrocephalus would contribute to the re-establishment of the SVZ niche leading to brain functions' improvement by enhancing neuronal production in human patients with hydrocephalus and ameliorating brain function. The in vivo reprogramming approach indicates that the in-situ conversion of cells into ependyma in its native microenvironment is feasible. Our in vivo approach allowed the exploration of the influence of the diseased environment of the hydrocephalic brain on the reprogramming process. Our data support the idea that functional maturation and integration of the reprogrammed cells in the native tissue can be achieved through our in vivo reprogramming method. Since the molecular program of multiciliogenesis is conserved across different organisms and tissues (Arbi et al, 2018), it is tempting to speculate that such a reprogramming approach could be applied to other diseases that impact ciliated tissues, such as the lung and the reproductive system. For instance, a rare genetic disorder affecting the mucociliary clearance mechanism, known as Reduced Generation of Multiple Motile Cilia (RGMC), has been linked to mutations in MCIDAS. Therefore, it would be highly interesting to investigate whether these conditions could be addressed in the future through a genetic treatment implementing McIdas' overexpression in the affected tissue.

Together, our analysis proposes MCIDAS and GEMC1 as potent reprogramming factors towards the ependymal cell lineage and demonstrate a proof of concept that restoring ependymal cells could contribute in managing hydrocephalus. Although this study yielded intriguing findings, our implementation holds certain limitations. In particular, our in vivo studies have been conducted to mouse models of hydrocephalus. Despite sharing a similar pathophysiology with the human disease, the anatomical and physiological characteristics of the mouse and human brain possess certain differences. Therefore, to validate the effectiveness of our approach, additional experiments in disease models closer to human pathophysiology will be required. The pig model demonstrates anatomical and physiological similarities to humans and replicates the pathophysiology of human hydrocephalus, exhibiting features such as ventriculomegaly, V/SVZ disruption, and neuroinflammation and could thus be a valuable model for examining the translational success of our approach to the human conditions (Garcia-Bonilla et al, 2023). In addition, since hydrocephalus in our study appeared during early developmental stages, our findings are particularly relevant to postnatal hydrocephalus. Therefore, their applicability to adult hydrocephalus will require further investigation using appropriate adult animal models. Although we achieved McIdas overexpression in the periventricular cells of the diseased brain through electroporation, such an approach cannot be used in humans. Therefore, alternative gene delivery methods compatible with therapeutic protocols, such as AAV gene delivery strategies in the human brain (Chamakioti et al, 2022; Au et al, 2022) should be established. Our study shows that regeneration of multiciliated ependymal cells can be succeeded in hydrocephalic mouse models while further experiments are needed to show the applicability of our findings in treating hydrocephalus in humans.

# Methods

**Reagents and tools table**

| Reagent/resource | Reference or source | Identifier or catalog number |
|---|---|---|
| **Experimental models** | | |
| GemC1-knockout mice | KOMP repository | |
| **Antibodies** | | |
| Chicken anti-GFP | Aves Laboratories | 1020 |
| Mouse anti-S100β | Sigma | S2532 |
| Rabbit anti-P73 | Abcam | ab40658 |
| Mouse anti-FoxJ1 | eBioscience | 14-9965 |
| Rabbit anti-pericentrin | Covance | PRB432C |
| Rabbit anti-GFAP | DakoCytomation | Z0334 |
| Mouse anti-acetylated α-tubulin | Sigma | T6793 |
| Rabbit anti-McIdas | Pefani et al, 2011 | |
| Mouse anti-centriolin | Santa Cruz | sc-365521 |
| Rabbit anti-Meig1 | Biorbyt | orb185566 |
| Rat anti-VCAM1 | BD Biosciences | 550547 |
| Mouse anti β-catenin | BD Biosciences | 610153 |
| Guinea pig anti-double-cortin | Millipore | AB2253 |

| Reagent/resource | Reference or source | Identifier or catalog number |
|---|---|---|
| **Chemicals, enzymes, and other reagents** | | |
| LPA | Avanti Polar Lipids | 857130 P |
| jet-OPTIMUS reagent | Polyplus | 101000051 |
| Mowiol 4-88 | Calbiochem | |
| Draq-5 | Biostatus | |
| Dapi | Sigma | |
| **Other** | | |
| Electroporator | Harvard Apparatus | ECM830 |
| Vibrating microtome | Leica | VT1000 S |

## Mouse knockout strain

C57/Bl6 mice carrying a knockout (KO) allele of the GemC1 gene were initially generated by the trans-NIH Knock-Out Mouse Project (KOMP) and obtained from the KOMP repository. The *GemC1*-KO allele was maintained on a mixed strain background. Inactivation of the *GemC1* gene was carried out using the "knockout-first" strategy, as previously described (Arbi et al, 2016; Lalioti et al, 2019b). This approach involves disrupting gene function by adding RNA processing signals without removing any of the targeted genes (Testa et al, 2004). The *GemC1*-KO allele was created by placing a lacZ-neo cassette between the second and third exon of the gene.

Mice were housed in the animal house of the University of Patras, where a standard light cycle of 12 h of light and 12 h of darkness was maintained. Continuous access to food and water was ensured at all times. To ensure animal welfare handling and environmental stressors like noise, vibrations, and strong odors were minimized to necessary levels. All animal-related procedures were conducted in strict accordance with EU directives and approved by the Veterinary Administrations of the Prefecture of Achaia, Greece, and the Research Ethics Committee of the University of Patras, Greece.

In all experiments, mice were used without regard to their gender. Experiments involving animal usage were conducted during the neonatal stage when the gender of the animals was not yet distinguishable.

## LPA preparation and injections

To generate an inducible mouse model of intracranial hemorrhage hydrocephalus, we performed intraventricular delivery of lysophosphatidic acid (LPA) in wild-type (WT) mice at postnatal day 5 (P5), following previously published works (Yung et al, 2011; Lummis et al, 2019).

Powdered 18:1 LPA (857130 P, Avanti Polar Lipids) was initially dissolved in methanol and subsequently vacuum-dried. LPA was then reconstituted in Hanks' balanced salt solution (HBSS 14065-049, Gibco) by sonicating in a water bath for 5 min to generate a 10 mM stock, as previously described (Yung et al, 2011; Lummis et al, 2019). Stock solution was stored at −20 °C until use. Using a glass microcapillary, 3 µl of the 10 mM LPA stock solution together

with Fast Green (0.1%, Sigma) were injected into the lateral ventricle of P5 mice upon hypothermia-anesthetization. Mice were sacrificed 2 days post LPA injections at P7 and were examined or further processed as described in the respective experiments.

## Plasmid constructs

For neonatal electroporation and mESCs transfection experiments mouse cDNAs for McIdas and GemC1 were cloned in pCAGGS-IRES-GFP vectors between SacI/EcoRV and NheI/SmaI restriction sites respectively, as described previously (Kyrousi et al, 2015).

## Neonatal electroporation

Neonatal electroporation was performed at P0-P1. Anesthesia was induced via hypothermia. Using a pulled glass microcapillary, 2 µg plasmid DNA together with Fast Green (0.1%, Sigma) were injected in the lateral ventricle of the mouse brains. Mice were subsequently electroporated with 5 pulses of 100 V for 50 ms each followed by intervals of 950 ms. An ECM830 Electroporator (Harvard Apparatus) and 7 mm platinum electrodes (BTX) were used. Following electroporation, mice were placed back to cages and left to grow for 4 or 6–14 days before sacrificed as described in the respective experiments.

## Ex vivo electroporation and organotypic brain slice cultures

For ex vivo electroporation, mice were sacrificed 2 days post LPA injections at P7. Mouse brains were dissected and placed in cold DMEM/F-12 medium (Gibco). Subsequently, 2 µg plasmid DNA of a pCAGGS-IRES-GFP (GFP) or pCAGGS-McIdas-IRES-GFP (McIdas) vector together with Fast Green (0.1%, Sigma) were injected in the lateral ventricle using a pulled glass microcapillary. Brains were then electroporated with 5 pulses applied at 100 V for 50 ms each at intervals of 950 ms. An ECM830 Electroporator (Harvard Apparatus) and 7-mm platinum electrodes (BTX) were used.

For organotypic brain slice cultures 250-µm thick coronal slices were obtained from the electroporated brains using a vibrating microtome (Leica VT1000 S). Consecutive slices containing the lateral ventricles were collected and placed in culturing medium [2% FBS (Gibco), 1% GlutaMAX Supplement (Gibco) and 1% penicillin–streptomycin (Gibco) in DMEM/F-12]. Subsequently, brain slices were placed onto 0.4 µm membrane inserts (Millipore) in six-well plates and cultured for 5 or 9 days in culturing medium, performing medium change every 3 days.

## Lentiviral production

A second generation packaging system was used to generate lentiviral expression particles as previously described (Arbi et al, 2016). The pLV-Dest-CAG lentiviral expression vector was used and was kindly provided by Dr. M. Gotz, Helmholtz Center, Munich. The cloning strategy for generating the lentiviral expression vectors for GFP and GemC1 with an N-terminal GFP tag (GFP-GemC1) was described elsewhere (Arbi et al, 2016).

McIdas with an N-terminal GFP tag (GFP-McIdas) was initially cloned into the KpnI/XhoI restriction sites of the

pENTR1AminusCmR vector. p73 with a C-terminal GFP tag (p73-GFP) was initially cloned into the SalI/NotI restriction sites of the pENTR1AminusCmR vector. An LR recombination reaction, using the Gateway LR Clonase II enzyme mix (Invitrogen), was performed between the attL-containing entry clone and the attR-containing destination pLV-Dest-CAG vector.

## Primary cultures of postnatal cortical astroglia and lentiviral infection

To establish primary cultures of postnatal cortical astroglia, the cerebral cortex of postnatal P5-P7 wild-type mice was dissected from the brain. Cells were subsequently mechanically dissociated and cultured under proliferating conditions for 7 days in astro medium containing DMEM/F-12 (Gibco), 10% FBS (Gibco), 2% B27 Supplement (Gibco), 1% GlutaMAX Supplement (Gibco), 1% penicillin–streptomycin (Gibco), epidermal growth factor (10 ng/ml) and basic fibroblast growth factor (10 ng/ml), as previously described (Heinrich et al, 2011).

For lentiviral infection, astroglial cells were plated onto poly-D-lysine pre-coated coverslips 7 days upon proliferation initiation and were left to attach and grow for 2–4 h. Cells were then transduced with concentrated lentiviral vectors. The infection mix was removed 24 h later and was replaced by astro medium which did not contain FBS, epidermal growth factor and basic fibroblast growth factor to favor differentiation. Cells were analyzed 7, 14 or 22 days later.

## Primary cultures of postnatal radial glial cells and lentiviral infection

To establish primary cultures of postnatal radial glial cells (pRGCs), the lateral walls of the lateral ventricles of P0-P1 mice were dissected, mechanically dissociated, and then plated onto poly-D-lysine pre-coated coverslips. pRGCs were cultured in proliferation medium [DMEM-high glucose (Gibco), 10% FBS (Gibco), and 1% penicillin/streptomycin (Gibco)] for 3 days (Paez-Gonzalez et al, 2011; Kyrousi et al, 2015; Lalioti et al, 2019b). Subsequently, pRGCs were infected with unconcentrated viral particles performing plate-spin infection. Proliferation medium was changed 24 h post infection and later replaced with differentiation medium [DMEM-high glucose (Gibco), 2% FBS (Gibco), and 1% penicillin/streptomycin (Gibco)] 48 h post infection. Cells were processed for analysis 5 or 15 days upon differentiation initiation.

## Cultures of mouse embryonic stem cells (mESCs) and transfection

Mouse embryonic stem cells were grown onto gelatin-coated (0.1% gelatin, Sigma) flasks in GMEM BHK-21 medium (Gibco), supplemented with 10% FBS (Biosera, Embryonic Stem Cells tested), 0.1 mM β-mercaptoethanol (Gibco), 2 mM L-glutamine (Sigma), 1 mM sodium pyruvate (Pan Biotech), 1% penicillin/streptomycin (Gibco), 1% MEM non-essential amino acids (Biosera) and 2.000 U/ml LIF (Millipore). Cells were passaged every other day and were mycoplasma tested.

Reverse mESCs transfection was performed using the jet-OPTIMUS reagent according to the manufacturer's instructions. Briefly, the transfection mixes containing DNA, jet-OPTIMUS

buffer and transfection reagent were prepared. Afterward, transfection mixes were placed on gelatin pre-coated glass coverslips and then mESCs were immediately seeded. Cells were cultured in the presence of LIF for 24 h. The following day the medium was changed (LIF withdrawal), and the cells were cultured in differentiating conditions for up to 6 days and were then processed for analysis.

## Video microscopy of ependymal cilia

For the study of ependymal cilia in primary cultures of reprogrammed postnatal cortical astroglia, cells were grown under proliferating conditions in astro medium for 7 days. Cells were then plated onto poly-D-lysine coated glass-bottom dishes (ibidi) and transduced with unconcentrated lentiviral vectors. 24 h post infection, medium was replaced by astro medium which did not contain FBS, epidermal growth factor and basic fibroblast growth factor to favor differentiation. Cells were used in video microscopy experiments 22 days later.

To perform video microscopy experiments in reprogrammed postnatal cortical astroglia, the culturing medium was replaced by live imaging medium containing MEM (Sigma), 20% FBS (Gibco), 4 mM L-glutamine (Sigma) and 1 mM sodium pyruvate (Pan Biotech). In addition, to track cilia-generated fluid flow 0.5-μm red fluorescent microbeads (1 μl/ml medium; Sigma) were added to the imaging medium. Video microscopy experiments were performed using an Olympus IX83 microscope equipped with a Hamamatsu Orca-Flash 4.0 sCMOS camera. During live imaging microscopy experiments cells were maintained at 37 °C and 5% $CO_2$. Videos were obtained using a 60× water immersion objective at 80 frames/s, while 500 frames were acquired for each field.

For the study of brain ependymal cilia, mice were sacrificed at P7–P15, brains were dissected in HBSS at room temperature and 200-μm thick coronal slices were obtained using a vibrating microtome. Slices containing the lateral ventricles were placed in DMEM-GlutaMAX (Gibco) media pre-warmed at 37 °C.

To perform video microscopy experiments, brain slices were placed onto glass-bottom dishes containing live imaging medium with 0.5-μm red fluorescent microbeads (1 μl/ml medium). Video recordings were obtained as described in the preceding paragraph.

For cilia beat frequency analysis, the ImageJ image processing program was used to firstly generate a graphical representation of cilia beating, in the form of a kymograph, as previously described by Drummond I. (Drummond, 2009). Cilia beat frequency for each multiciliated cell was then calculated with the correlation of the time scale of the video recording with the kymograph, following the method described by Francis R. and Lo C. (Francis and Lo, 2013).

To track the trajectory of the moving microbeads the MTrackJ Plugin of the ImageJ program was used. In short, the point selection tool was used to manually mark the position of each red fluorescent microbead in each frame. Consecutive marks on the corresponding video recordings created the trajectory of the microbeads.

## Whole-mount dissection

SVZ whole mounts were obtained as previously described (Lalioti et al, 2019b). Briefly, the brain was isolated and the electroporated lateral ventricle was carefully separated from the posterior region of

the telencephalon, followed by the extraction of the hippocampus and septum. Fixation of the dissected lateral wall was performed in 4% PFA/0.1% Triton X-100 at 4 °C overnight. Staining was then performed (see below, immunofluorescence) and the lateral wall was further dissected from the surrounding brain parenchyma, resulting in tissue slivers with a thickness of 200–300 micrometers. These slivers were then affixed to a glass slide using Mowiol 4-88 (Calbiochem) mounting media and covered with a coverslip.

## Immunofluorescence

For immunofluorescence, cultured cells were fixed for 10 min with 4% PFA (or 10 min PFA 4% followed by 10 min fixation in Methanol where stated) and incubated in blocking solution, for 1 h at room temperature. Postnatal cortical astroglial cells were incubated in blocking solution containing 5% FBS, 3% BSA, 0.1% Triton X-100 in 1× PBS. pRGCs and mESCs were incubated in blocking solution containing 5% Normal Goat Serum (Jackson Immunoresearch) and 0.3% Triton X-100 in 1× PBS. Cells were incubated with primary antibodies in blocking solution at 4 °C, overnight (Kalogeropoulou et al, 2022).

Brains dissected from newborn mice (up to P7) were fixed overnight with 4% PFA. Subsequently, brains were rinsed with 1× PBS and cryopreserved with incubation in 30% sucrose in 1× PBS for ~48 h. Brains were then frozen in 7.5% gelatin supplemented with 15% sucrose and sectioned at 12 μm using a CM1850 Leica cryostat. Brain coronal cryosections were postfixed with 4% PFA for 10 min, treated with 0.3% Triton X-100 in 1× PBS for 5 min, and incubated in blocking solution containing 5% FBS, 3% BSA, 0.1% Triton X-100 in 1× PBS, for 1–3 h. Samples were incubated with primary antibodies in blocking solution at 4 °C, overnight.

For immunofluorescence of cultured brain slices, membrane inserts were removed, rinsed in 1× PBS, and fixed with 4% PFA at 4 °C, overnight. Membrane inserts were then subjected to consecutive washes in 1× PBS (5 min), 0.1% Triton X-100 in 1× PBS (5 min) and 0.5% Triton X-100 in 1× PBS (20 min once) and were incubated in blocking solution containing 1% FBS, 0.1% Triton X-100 in 1× PBS for 2–3 h at room temperature. Samples were incubated with primary antibodies in blocking solution at 4 °C, overnight.

Immunofluorescence of SVZ whole mounts was performed as previously described (Mirzadeh et al, 2008, 2010). Briefly, whole mounts were incubated for 1–2 h at room temperature in blocking solution, containing 10% normal goat serum in 0.1 M PBS with 0.5–2% Triton X-100. Samples were incubated with primary antibodies in blocking solution at 4 °C for 24–48 h.

The following primary antibodies were used: chicken anti-GFP (1020, Aves Laboratories, 1:500 for brain sections; 1:1000 for cells, 1:250 for whole mounts), mouse anti-S100β (S2532, Sigma, 1:250), rabbit anti-P73 (ab40658, Abcam, 1:300), mouse anti-FoxJ1 (14-9965, eBioscience, 1:500), rabbit anti-pericentrin (PRB432C, Covance, 1:750 for brain sections; 1:1500 for cells), rabbit anti-GFAP (Z0334, DakoCytomation, 1:1000 for cells, 1:500 for whole mounts), mouse anti-acetylated α-tubulin (T6793, Sigma, 1:1000 for brain sections; 1:1500 for cells), rabbit anti-McIdas ((Pefani et al, 2011), 1:250), mouse anti-centriolin (sc-365521, Santa Cruz, 1:100, cells' fixation in PFA-Methanol), rabbit anti-Meig1 (orb185566, Biorbyt, 1:100), rat anti-VCAM1 (550547, BD

Biosciences, 1:100), mouse anti β-catenin (610153, BD Biosciences, 1:500), guinea pig anti-double-cortin (AB2253, Millipore, 1:500).

The following Alexa Fluor-labeled secondary antibodies (Invitrogen) were used in the respective blocking solution (1:1000) for 1 h (cryosections and cells) at room temperature or overnight (membrane inserts) at 4 °C: Alexa Fluor 488 goat anti-chicken, Alexa Fluor 568 goat anti-mouse IgG1, Alexa Fluor 647 donkey anti-rabbit, Alexa Fluor 568 goat anti-rabbit, Alexa Fluor 647 goat anti-mouse IgG1, Alexa Fluor 568 goat anti-mouse, Alexa Fluor 568 goat anti-rabbit, Alexa Fluor 647 donkey anti-mouse, Alexa Fluor 568 goat anti-guinea pig and Alexa Fluor 568 goat anti-rat. DNA was stained either with Draq-5 (1:1000, Biostatus) or DAPI (1:1500, Sigma). Samples were mounted in Mowiol 4-88 (Calbiochem).

## Microscopy and data analysis

Images of fluorescent samples were acquired with a confocal Leica TCS SP5 and a Leica TCS SP8 system equipped with a fluorescent Leica DMI600B or Leica DMi8 microscope, respectively. In all, ×20, ×40, and ×63 lenses were used. Digital images were processed with Adobe Photoshop and the ImageJ image processing program. For still images related to video microscopy experiments, individual frames were cropped and adjusted for brightness and contrast in ImageJ.

Quantification of cells on overexpressed brains from in vivo experiments was performed on cryosections, vibratome sections or whole mounts. In all instances, each brain region containing electroporated cells was imaged and quantified per frame. All quantifications are expressed as percentages of the total cell count. Quantification of radial glial cells, cortical astrocytes, or mESCs from ex vivo experiments was performed in at least two independent experiments, with a minimum of 15 different images analyzed for each condition in each experiment. Fluorescent intensity per nucleus was measured using the open-source platform ImageJ.

The number of independent experiments performed for each analysis is mentioned in the corresponding figure legend. Normality was tested for each experiment using the Shapiro–Wilk test to

### The paper explained

#### Problem

Hydrocephalus is a condition where fluid builds up in the brain cavities, leading to serious health issues. Standard treatment involves surgery, which is risky and often fails over time. Ependymal cells have multiple tiny hair-like structures (cilia) on their surface to circulate the fluid, and their impairment may lead to hydrocephalus.

#### Results

We showed that GemC1 and McIdas, which are known regulators of multiciliated ependymal cell fate determination, can instruct different cell types to differentiate into ependymal cells. We further managed to convert the brain cells of hydrocephalic animal models into ependymal cells.

#### Impact

Our findings suggest a new therapeutic intervention aiming at the regeneration of damaged ependymal cells in human hydrocephalus.

determine if the sample data came from a normally distributed population. Since our data did not follow a normal distribution, statistical analysis was performed with the nonparametric two-tailed Mann–Whitney test. Significant differences in medians were determined by $P$ value: $P < 0.05$ (*$P < 0.05$, **$P < 0.01$, ***$P < 0.001$). Each graph displayed the median ± interquartile range (IQR). All statistical analyses and graph preparation were performed in GraphPad Prism 6.

The study did not incorporate blinding or randomization, and all samples were included without applying specific inclusion or exclusion criteria. These choices were guided by the exploratory nature of the research.

## Data availability

This study includes no data deposited in external repositories.

The source data of this paper are collected in the following database record: biostudies:S-SCDT-10_1038-S44321-024-00156-5.

## Peer review information

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

## Acknowledgements

The authors thank the Advanced Light Microscopy Facility of the Medical School at the University of Patras and the Experimental Animal Facility of the University of Patras for support with experiments. We are also grateful to Andriana Charalampopoulou and members from Professor's Benedikt Berninger laboratory for their assistance in astrocytic reprogramming experiments. The authors also thank Dr. Christina Kyrousi and members of our laboratories for helpful discussions and Nikolaos Giakoumakis for his assistance with whole-mount images processing. This study was supported by the Hellenic Foundation for Research and Innovation (H.F.R.I.) under the "2nd Call for H.F.R.I. Research Projects to support Faculty Members & Researchers" (Project Number: 2735), the Hydrocephalus Association USA, Fondation Santé and the European Union's Horizon Europe (2021-2027), "ESPERANCE" ERA Chair program (GA 101087215) to ST, and the State Scholarships Foundation (IKY) to KK. The publication fees of this manuscript have been financed by the Research Council of the University of Patras.

## Author contributions

**Konstantina Kaplani**: Conceptualization; Funding acquisition; Investigation; Visualization; Methodology; Writing—original draft; Project administration; Writing—review and editing. **Maria-Eleni Lalioti**: Conceptualization; Investigation; Visualization; Methodology; Project administration; Writing—review and editing. **Styliani Vassalou**: Investigation; Visualization; Methodology. **Georgia Lokka**: Methodology. **Evangelia Parlapani**: Investigation; Visualization; Methodology. **Georgios Kritikos**: Methodology. **Zoi Lygerou**: Writing—review and editing. **Stavros Taraviras**: Conceptualization; Supervision; Funding acquisition; Project administration; Writing—review and editing.

Source data underlying figure panels in this paper may have individual authorship assigned. Where available, figure panel/source data authorship is listed in the following database record: biostudies:S-SCDT-10_1038-S44321-024-00156-5.

## Disclosure and competing interests statement

Zoi Lygerou is an *EMBO* Council member. This has no bearing on the editorial consideration of this article for publication. The remaining authors declare no competing interests.

# Expanded View Figures

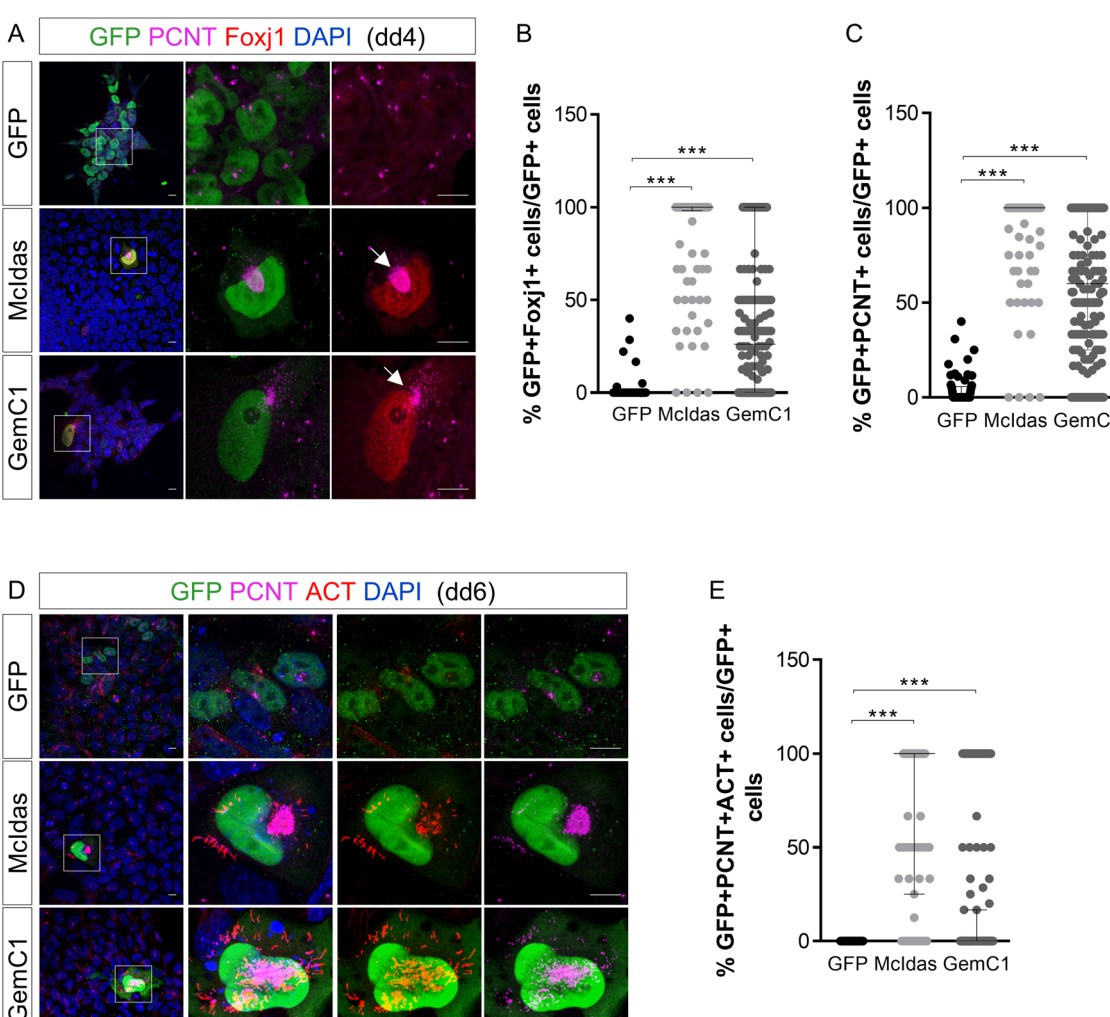

**Figure EV1. Mouse embryonic stem cells programming into ependymal cells.**

(A) Mouse embryonic stem cells (mESCs) were transfected with GFP, GFP-McIdas (McIdas) or GFP-GemC1 (GemC1) expressing vectors and subsequently cultured under differentiating conditions. Immunofluorescence was performed with antibodies against GFP (green), FoxJ1 (red) and Pericentrin (PCNT, magenta) at differentiation day 4. The arrow points to multiple basal bodies in McIdas and GemC1 transfected cells which co-express the ependymal marker FoxJ1. Higher magnification of the boxed regions is shown in the right panels. (B, C) The graphs present the percentage of transfected mESCs that express FoxJ1 over the total number of the transfected cells (B) and the percentage of transfected cells which display multiple basal bodies based on the accumulation of PCNT signal. (C) Data are presented as the median ±IQR of two independent experiments. Statistical significance was determined using the nonparametric two-tailed Mann–Whitney test (***$P < 0.0001$). (D) GFP (green), PCNT (magenta) and acetylated α-tubulin (ACT red) immunostaining in transfected mESCs at differentiation day 6. McIdas and GemC1 transfected mESCs present multiple basal bodies and cilia. (E) The graph shows the percentage of transfected cells with multiple basal bodies (PCNT accumulation) and multiple cilia (ACT labeled cilia) over the total number of the transfected cells. Data are presented as the median ±IQR of two independent experiments. Statistical significance was determined using the nonparametric two-tailed Mann–Whitney test (***$P < 0.0001$). DNA was stained with DAPI (blue). Scale bars, 10 μm. dd differentiation day.

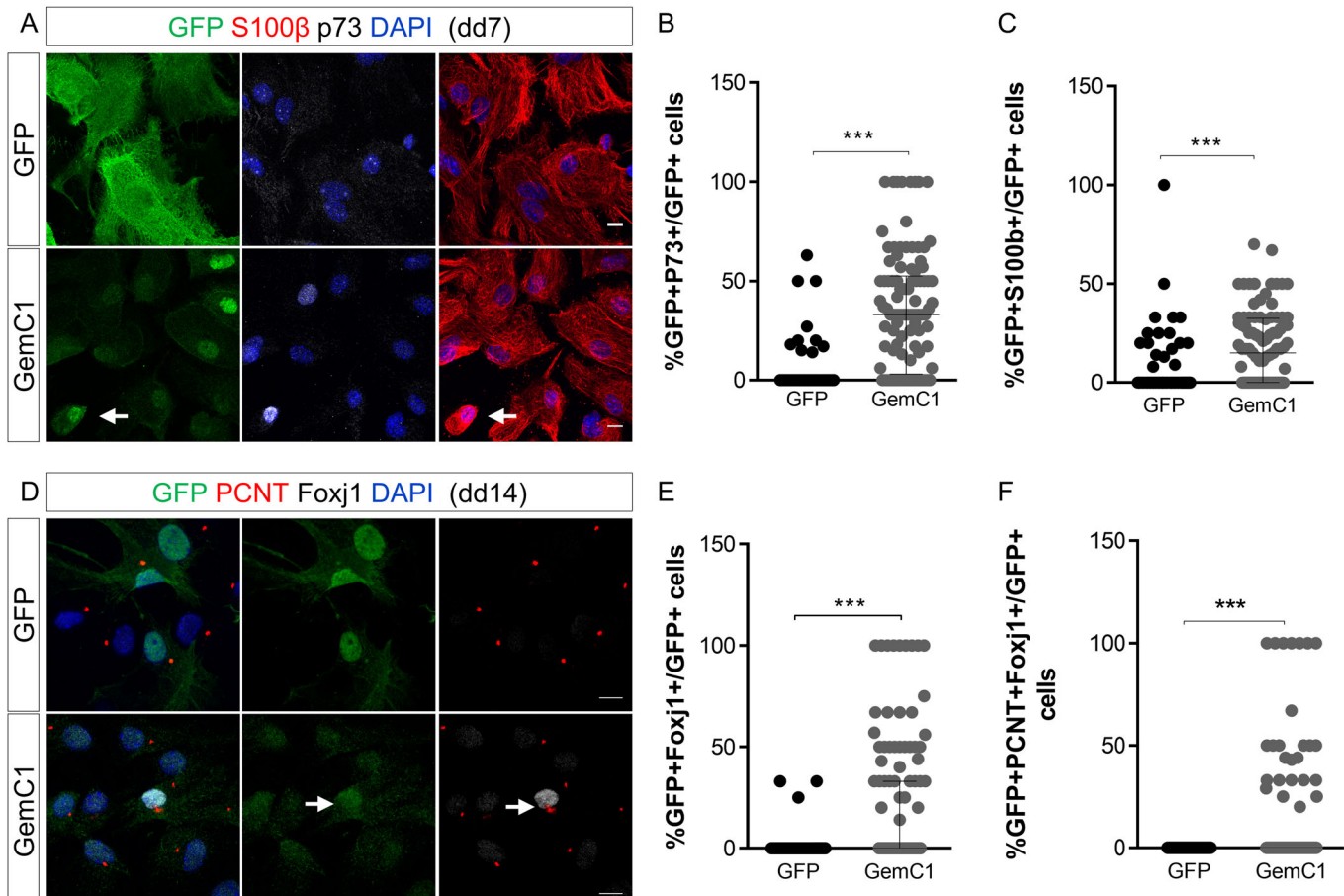

**Figure EV2. GemC1 drives the commitment of murine cortical astrocytes towards the ependymal cell lineage.**

(A) Cortical astrocytes were transduced with lentiviruses encoding GFP or GFP-GemC1 (GemC1). Immunofluorescence experiments were performed at day 7 of differentiation with specific antibodies against GFP (green) to mark the infected cells, P73 (gray) and S100β (red). Arrow point to a P73+ infected cell, committed to the ependymal lineage. (B, C) Graph depicting the percentage of infected cells which express P73 over the total number of infected cells (B). The graph shows the percentage of the infected cells which display S100β staining around their cell body (C). Three independent experiments were analyzed. Data are presented as the median ±IQR. Statistical significance was determined using the nonparametric two-tailed Mann–Whitney test (***P < 0.0001). (D) Astrocytes were immunostained against GFP (green), FoxJ1 (gray) and pericentrin (PCNT red) at differentiation day 14. The arrow points to a GemC1-infected cell which expresses FoxJ1 and possesses multiple basal bodies. (E, F) Graph presenting the percentage of the infected cells that express FoxJ1 over the total number of the infected cells. (E) The graph shows the percentage of infected cells which express the ependymal marker FoxJ1 and display accumulation of PCNT signal (F). Data are presented as the median ±IQR from three independent experiments. Statistical significance was determined using the nonparametric two-tailed Mann–Whitney test (***P < 0.0001). DNA was stained with DAPI (blue). Scale bars, 10 μm. dd differentiation day.

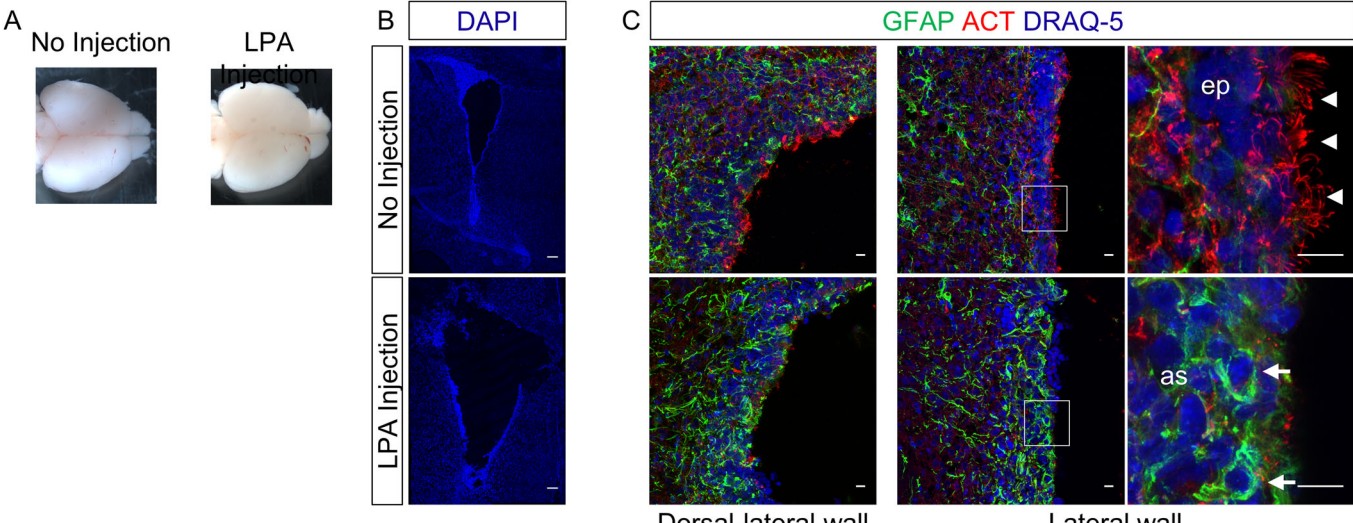

**Figure EV3. Lysophosphatidic Acid (LPA) administration causes hydrocephalus in early postnatal stages in mice.**

(A) Top views of Postnatal day 7 (P7) wild-type mouse brains that either received or not injection of LPA at the lateral ventricle at P5. No differences on the size of the brain were observed macroscopically. (B) DAPI staining on coronal brain sections from P7 mouse brains reveals the dilation of the lateral ventricles after LPA injections. Scale bar, 100 μm. (C) Immunofluorescence for Glial Fibrillary Acidic Protein (GFAP, green) which labels astrocytes and acetylated α-tubulin (ACT, red), a marker of ciliary axonemes on coronal sections from P7 mouse brains at the dorsal-lateral and lateral regions of the lateral walls. Arrowheads point to multiple cilia in control ependymal cells (ep). Arrows show GFAP-positive astrocytes (as) in the lateral wall of LPA-injected brains, where ciliary disruption is observed. Higher magnification of the boxed regions is shown in the right panels. Scale bars, 10 μm. DNA was stained with DAPI or Draq-5 (blue).

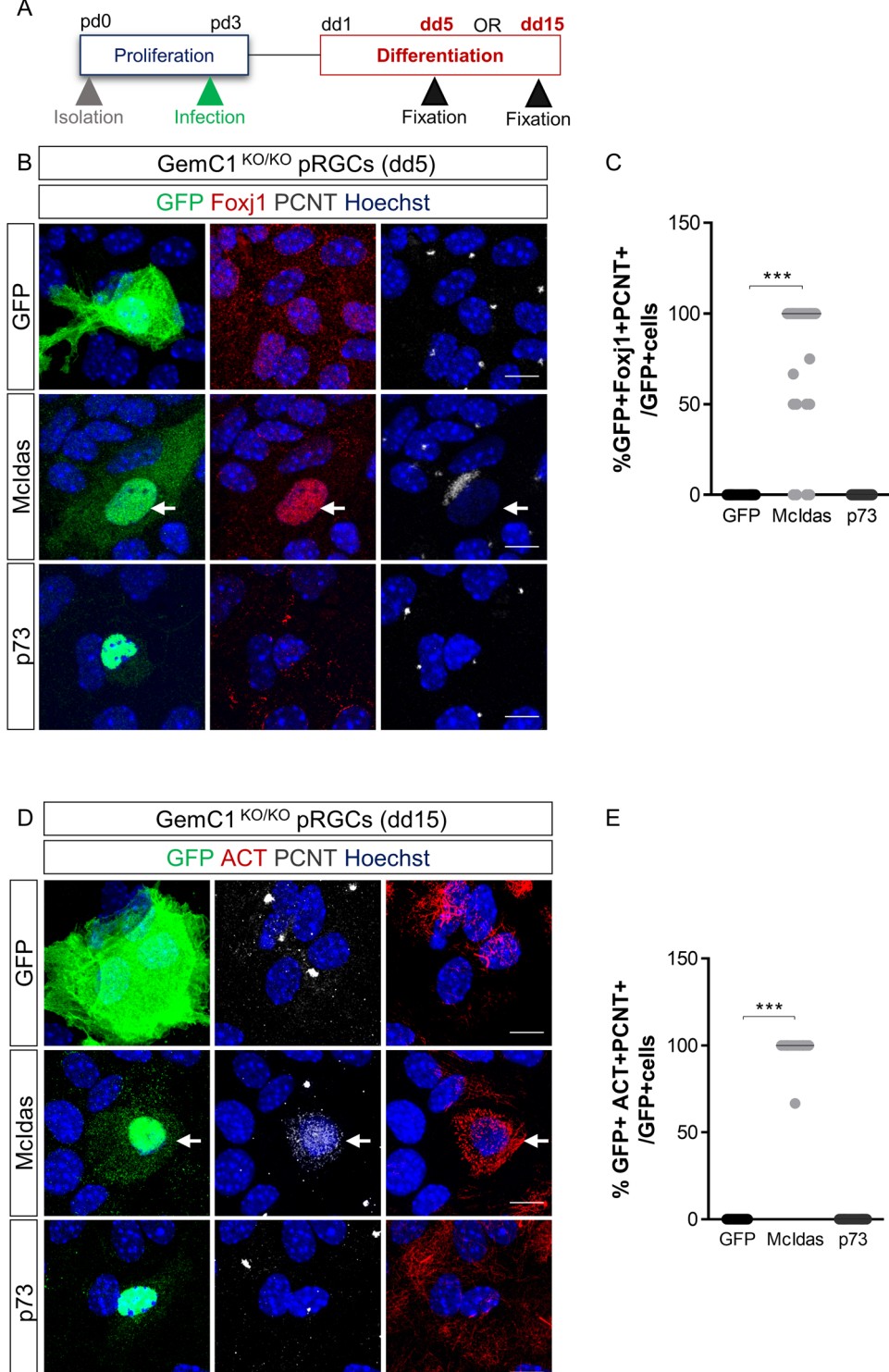

◄ **Figure EV4. Ex vivo reprogramming of radial glial cells to ependyma can be achieved through McIdas ectopic expression.**

(A) Schematic representation of experimental procedures for (B–E). Postnatal radial glial cells (pRGCs) were isolated from GemC1-knockout newborn mice, cultured and infected with lentiviruses expressing a GFP-McIdas (McIdas) or a GFP-P73 (P73) fusion protein, while GFP alone was used as a control. Cells were then cultured under differentiating conditions and analyzed at indicated time points. (B) Transduced radial glial cells were co-stained with antibodies against GFP (green) to mark infected cells, FoxJ1 (red) and pericentrin (PCNT, gray) 5 days after the initiation of differentiation. The arrow indicates the accumulation of PCNT signal in FoxJ1 expressing cells upon McIdas ectopic expression. (C) The graph presents the percentage of infected cells which express FoxJ1 and display multiple basal bodies (accumulation of PCNT signal). Three independent experiments were analyzed. Data are presented as the median ±IQR. Statistical significance was determined using the nonparametric two-tailed Mann–Whitney test (***$P < 0.0001$). (D) Radial glial cells infected with GFP, McIdas, or P73 lentiviruses were labeled with antibodies against GFP (green), pericentrin (PCNT, gray) and acetylated α-tubulin (ACT, red) to detect mature multiciliated cells at differentiation day 15 (arrow). (E) The percentage of the infected cells which displayed multiple basal bodies, based on PCNT staining and simultaneously multiple cilia, based on ACT staining, was analyzed. Data are presented as the median ±IQR of three independent experiments. Statistical significance was determined using the nonparametric two-tailed Mann–Whitney test (***$P < 0.0001$). DNA was stained with Hoechst (blue). Scale bars, 10 μm.

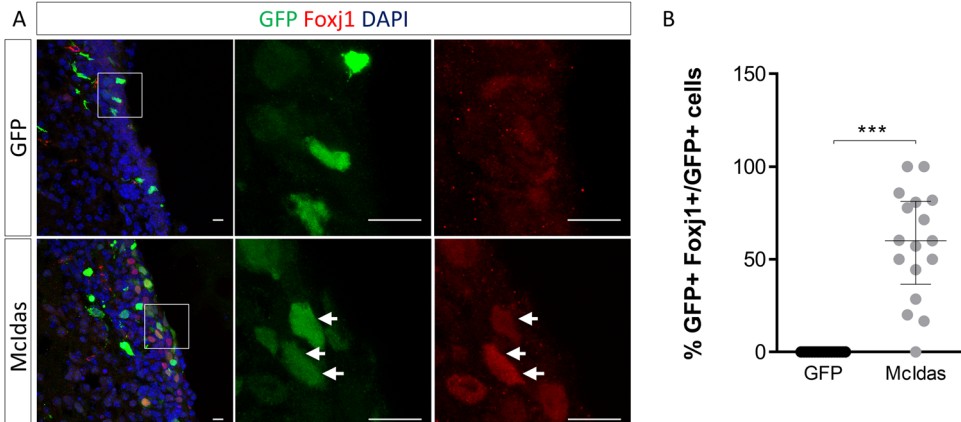

**Figure EV5.  McIdas drives the commitment to the ependymal cell lineage in a genetic mouse model for hydrocephalus.**

(**A**) Subventricular zone electroporation was conducted at postnatal day 1 (P1) GemC1-knockout mice with plasmids encoding GFP, or GFP-McIdas (McIdas). Coronal brain sections were stained with antibodies against GFP (green) to mark the electroporated cells and FoxJ1 (red), a marker of committed ependymal cells, 4 days post electroporation. Arrows point to McIdas-electroporated cells expressing FoxJ1. Higher magnification of the boxed region is shown in the right panel. (**B**) Graph depicting the percentage of electroporated cells that express FoxJ1 over the total number of the electroporated cells. Two independent experiments were analyzed. Data are presented as the median ±IQR. Statistical significance was determined using the nonparametric two-tailed Mann–Whitney test (***$P < 0.0001$). DNA was stained with DAPI (blue). Scale bars, 10 μm.

                                      