## [Peer Review File · EMBO Molecular Medicine]

Investigating ependymal cell lineage reprogramming as a therapeutic intervention for hydrocephalus

Konstantina Kaplani, Maria-Eleni Lalioti, Stella Vassalou, Georgia Lokka, Evangelia Parlapani, Georgios Kritikos, Zoi Lygerou, and Stavros Taraviras

Corresponding author: Stavros Taraviras (taraviras@med.upatras.gr)

Review Timeline:

Submission Date:	23rd Nov 23
Editorial Decision:	15th Jan 24
Revision Received:	14th Jul 24
Editorial Decision:	5th Aug 24
Revision Received:	22nd Sep 24
Accepted:	8th Oct 24

Editor: Lise Roth

Transaction Report:

15th Jan 2024

Dear Prof. Taraviras,

Thank you for the submission of your manuscript to EMBO Molecular Medicine, and please accept my apologies for the delay in getting back to, as during this time of the year it is challenging to secure referees and get reports in a timely manner. We have now heard back from the referees who agreed to evaluate your manuscript. As you will see below, the reviewers raise substantial concerns on your work, which unfortunately preclude its publication in EMBO Molecular Medicine in its current form. The reviewers find that the question addressed by the study is of potential interest, however, they remain unconvinced that some of the major conclusions are sufficiently supported by the data.

If you feel you can satisfactorily address the concerns listed by the referees, you may wish to submit a revised version of your manuscript. Please attach a covering letter giving details of the way in which you have handled each of the points raised by the referees. A revised manuscript will once again be subject to review, and we cannot guarantee at this stage that the eventual outcome will be favorable.

We are expecting your revised manuscript within three to six months, if you anticipate any delay, please contact us.

We require:

- 1) A .docx formatted version of the manuscript text (including legends for main figures, EV figures and tables). Please make sure that the changes are highlighted to be clearly visible.
- 2) Individual production quality figure files as .eps, .tif, .jpg (one file per figure). For guidance, download the 'Figure Guide PDF' (<https://www.embopress.org/page/journal/17574684/authorguide#figureformat>).
- 3) At EMBO Press we ask authors to provide source data for the main figures. Our source data coordinator will contact you to discuss which figure panels we would need source data for and will also provide you with helpful tips on how to upload and organize the files.
- 4) A .docx formatted letter INCLUDING the reviewers' reports and your detailed point-by-point responses to their comments. As part of the EMBO Press transparent editorial process, the point-by-point response is part of the Review Process File (RPF), which will be published alongside your paper.
- 5) A complete author checklist, which you can download from our author guidelines (<https://www.embopress.org/page/journal/17574684/authorguide#submissionofrevisions>). Please insert information in the checklist that is also reflected in the manuscript. The completed author checklist will also be part of the RPF.
- 6) It is mandatory to include a 'Data Availability' section after the Materials and Methods. Before submitting your revision, primary datasets produced in this study need to be deposited in an appropriate public database, and the accession numbers and database listed under 'Data Availability'. Please remember to provide a reviewer password if the datasets are not yet public (see <https://www.embopress.org/page/journal/17574684/authorguide#dataavailability>). In case you have no data that requires deposition in a public database, please state so in this section. Note that the Data Availability Section is restricted to new primary data that are part of this study.
- 7) For data quantification: please specify the name of the statistical test used to generate error bars and P values, the number (n) of independent experiments (specify technical or biological replicates) underlying each data point and the test used to calculate p-values in each figure legend. The figure legends should contain a basic description of n, P and the test applied. Graphs must include a description of the bars and the error bars (s.d., s.e.m.). Please provide exact p values.
- 8) We replaced Supplementary Information with Expanded View (EV) Figures and Tables that are collapsible/expandable online. A maximum of 5 EV Figures can be typeset. EV Figures should be cited as 'Figure EV1, Figure EV2' etc... in the text and their respective legends should be included in the main text after the legends of regular figures.
- For the figures that you do NOT wish to display as Expanded View figures, they should be bundled together with their legends in a single PDF file called *Appendix*, which should start with a short Table of Content. Appendix figures should be referred to in

the main text as: "Appendix Figure S1, Appendix Figure S2" etc.

9) The paper explained: EMBO Molecular Medicine articles are accompanied by a summary of the articles to emphasize the major findings in the paper and their medical implications for the non-specialist reader. Please provide a draft summary of your article highlighting

10) For more information: There is space at the end of each article to list relevant web links for further consultation by our readers. Could you identify some relevant ones and provide such information as well? Some examples are patient associations, relevant databases, OMIM/proteins/genes links, author's websites, etc...

11) Author contributions: CRediT has replaced the traditional author contributions section because it offers a systematic machine readable author contributions format that allows for more effective research assessment. Please remove the Authors Contributions from the manuscript and use the free text boxes beneath each contributing author's name in our system to add specific details on the author's contribution. More information is available in our guide to authors.

12) Disclosure statement and competing interests: We updated our journal's competing interests policy in January 2022 and request authors to consider both actual and perceived competing interests. Please review the policy <https://www.embopress.org/competing-interests> and update your competing interests if necessary.

13) Every published paper now includes a 'Synopsis' to further enhance discoverability. Synopses are displayed on the journal webpage and are freely accessible to all readers. They include a short stand first (maximum of 300 characters, including space) as well as 2-5 one-sentences bullet points that summarizes the paper. Please write the bullet points to summarize the key NEW findings. They should be designed to be complementary to the abstract - i.e. not repeat the same text. We encourage inclusion of key acronyms and quantitative information (maximum of 30 words / bullet point). Please use the passive voice. Please attach these in a separate file or send them by email, we will incorporate them accordingly.

14) As part of the EMBO Publications transparent editorial process initiative (see our Editorial at <http://embomolmed.embopress.org/content/2/9/329>), EMBO Molecular Medicine will publish online a Review Process File (RPF) to accompany accepted manuscripts.

In the event of acceptance, this file will be published in conjunction with your paper and will include the anonymous referee reports, your point-by-point response and all pertinent correspondence relating to the manuscript. Let us know whether you agree with the publication of the RPF and as here, if you want to remove or not any figures from it prior to publication. Please note that the Authors checklist will be published at the end of the RPF.

I look forward to receiving your revised manuscript.

Yours sincerely,

Lise Roth

**** Reviewer's comments ****

Referee #1 (Comments on Novelty/Model System for Author):

There were both in vitro (ex vivo) and in vivo models used, including primary cell cultures of early postnatal astrocytes and radial glia cells and early postnatal mice. To improve the relevance of the findings, the radial glia primary cultures should be isolated from early adult mice and selected in vivo experiments should be replicated in young adult mice.

Referee #1 (Remarks for Author):

This manuscript identifies GemC1 and Mclidas, two geminin family proteins, as factors that can reprogram astrocytes and Embryonic Stem Cells (ESC) to ependymal cells. The authors use a combination of in vitro and in vivo approaches to directly cell reprogram cells to functional ependymal cells. Some of the experiments are conducted in the context of the hydrocephalus, providing a pre-clinical relevance to the data. The results are convincing and consistent and provide evidence that GemC1 and/or Mclidas may potentially be used in a cell-replacement therapies of the hydrocephalus. However, the applicability of the presented results should be discussed more critically and there are important experiments required to characterize the astrocyte (or ESC) to ependyma conversion. When these experiments are added, I am confident that the manuscript would be a valuable addition to the field of cell reprogramming and hydrocephalus.

Major specific comments:

1. The segment on the statistical analysis in the methods needs to be expanded and the data must be presented correctly. The authors state that they used non-parametric statistical tests (i.e., Mann-Whitney test), however, they present all data as mean {plus minus} SEM. This is incorrect. Non-parametric tests do not compare means. Data in all graphs that were analysed by non-parametric tests should use median {plus minus} IQR or similar range indicator. The section on the statistical analysis should also indicate if the normality of data distribution was tested and if yes, why were not parametric test used. It should be made clear what statistical software was used.
2. Conversion of astrocytes to ependyma needs to be characterised in more detail. The authors use reduced or altered expression of S100b as marker of diminished astrocytic cell identity. This is not sufficient. The authors should combine S100b, GFAP and astrocytic cell morphology to demonstrate that the GemC1 or Mclidas over-expressing cells loose astrocytic identity. This can be done either in vitro or in vivo.
3. More thorough characterisation of ependymal reprogramming. The authors use P73, PCNT or FoxJ1 as markers of ependymal identity and the successful reprogramming. This is not sufficient. The authors should use other ependymal markers to characterize the cell conversion more accurately and thoroughly. These markers may include Fam183b, Lrcc1 or Meig1 (MacDonald et al., Front Cell Neurosci 2021) and should be used to characterize reprogrammed astrocytes or ESC, preferably in vivo.
4. Why is GemC1 less effective than Mclidas in reprogramming? In all presented experiments, GemC1 is less effective in reprogramming cells to the ependymal cell fate. But why? This should be addressed. One approach could be testing how much expression of Mclidas is in astrocytes, for example, where GemC1 was over-expressed.
5. Can astrocyte to ependyma reprogramming mitigate the hydrocephalus? In the GemC1 KO mice, the author should determine if the Mclidas-driven reprogramming reduces the cross-section or volume of the lateral ventricles as a readout of resolved or reduced hydrocephalus.
6. The effects of GemC1 or Mclidas reprogramming on the structure of V-SVZ pinwheels needs to be quantified more thoroughly. Figure 6 shows only representative images. It should show the proportion of infected (GFP+) cells that were able to form pinwheels in the V-SVZ wholemounts. In addition, the representative images of pinwheels should be shown without the cell outlines, which obstruct the structures.

Minor specific comments

A. Figure 3A seems to have a high background for P73 staining. Some of the P73+ cells in the panel A of LPA-Mclidas appear to show background and not true P73 signal. Can a better representative image be used?

B. The representative pictures in Figure S2 are not very convincing, especially the S100b staining used to claim that the reprogrammed astrocytes change the location of S100b in the cell compartments. Better representative images should be provided.

C. The authors should tone down the direct medical applicability of their research both in the abstract and in the discussion.

D. In the discussion, the authors should emphasize and highlight that their experiment were all done in early postnatal animals only and thus the applicability of their findings is limited only to postnatal and not to adult hydrocephalus.

E. In the discussion, the authors state that "additional experiments in disease models closer to human pathophysiology will be required". This is a vague statement. The authors should specify what disease models closer to humans they had in mind.

Referee #2 (Comments on Novelty/Model System for Author):

Fate mapping studies and deeper molecular characterization of the intermediate stages could improve a study that is nevertheless novel and well executed.

Referee #2 (Remarks for Author):

In their manuscript titled 'Ependymal cell lineage reprogramming as a therapeutic approach for hydrocephalus,' Kaplani and colleagues investigate the potential therapeutic strategy of reprogramming towards ependymal cell lineage for the treatment of hydrocephalus. The manuscript focuses on the study of GemC1 and MclDas as 'reprogramming factors', both of which have been extensively characterized in previous studies as crucial factors in establishing the ependymal cell compartment in the adult brain by the same authors. The authors first demonstrate a role for GemC1 and MclDas overexpression in directing the differentiation of mouse embryonic stem cells and immature astrocytes toward an ependymal cell fate in vitro. Subsequently, the authors evaluate the effects of MclDas overexpression in vivo, where they observe a significant promotion of ependymal cell regeneration in hydrocephalus mouse models, including an ependymal cell-deficient genetic background (GemC1-KO mice). Collectively, the study introduces a novel genetic approach for inducing ependymal cell identity within resident neural cells by regulating MclDas expression, which may hold implications for conditions related to ependymal cells, such as hydrocephalus.

The study presents compelling evidence for the role of GemC1 and, most notably, MclDas in promoting ependymal cell identity across various cellular contexts.

However, it is important to note that the term 'direct cellular reprogramming' does not precisely capture the experimental outcomes demonstrated in this study. While the experiments with mouse embryonic stem cells (mESCs) and cultured astrocytes (Fig. 1 and 2) clearly indicate the promotion of stem cell differentiation towards an ependymal fate, they do not unequivocally establish direct reprogramming. Additionally, considering the presence of neural stem cells in the ventricular wall, the in vivo experiments (Fig. 3-6) could also be interpreted as the differentiation of these progenitors towards an ependymal fate, and not as a direct change in fate from a postmitotic cell of another lineage i.e. astrocytes, as the authors indicate. Therefore, it is essential to describe the observed events more accurately, such as 'differentiation towards an ependymal fate.' It would be beneficial to address these possibilities in the discussion section, providing a more comprehensive analysis of the different potential cellular sources and differentiation mechanisms.

Furthermore, several experiments presented in the manuscript could potentially be interpreted as promoting the survival of ependymal cells. For instance, the in vitro experiments and the LPA model data should be considered in this context (there seems to be a large presence of GFP-negative P73 cells in the LPA model). It would be advisable to directly discuss and present this possibility as one of the potential outcomes, especially in the absence of direct fate mapping of the starting cell populations.

The experiments in GemC1 knockout mice are particularly interesting and the reconstitution of functional ciliated cells and ventricular pinwheel structures elicited by MclDas overexpression is remarkable. In Figure 5 and associated movies, the authors provide evidence for cilia function. However, it appears that most of the ciliated cells are FITC/EGFP negative. If so, the authors should clarify what is the origin of those cells and otherwise provide further evidence for a direct link between the functional cells and expression of MclDas.

The authors should consider/discuss the possibility that GemC1/MclDas expression alters the expression pattern of the EGFP reporter gene itself, thus biasing the studied populations, as has been reported for e.g. NeuroD1 overexpression constructs. The experiments in the ependymal-deficient mouse model are critical in excluding this possible confounding factor, but it could affect the interpretations of the other experiments.

Minor comments:

- Present replicate numbers directly in the graph as individual data points instead of only the summary bar graph

- Details on how the different quantifications were performed are missing. Please explain in detail how the in vitro and particularly in vivo experiments were quantified.

Overall, this study presents interesting findings regarding ependymal cell regeneration in the context of hydrocephaly. Clarifying the terminology and discussing the evidence in the context of cellular differentiation will enhance the scientific rigor and completeness of the manuscript in the absence of labor-intensive fate mapping studies supporting direct reprogramming.

Referee #3 (Comments on Novelty/Model System for Author):

This manuscript by Kaplani et al. provides overall convincing evidence that forced expression of *Mcl*das and to lesser degree *GemC1* induces ependymal cell features in mouse embryonic stem cells, primary astrocytes, isolated radial glia in culture as well as presumable non-ependymal cells in vivo in pharmacological and genetic models of hydrocephalus. With regard to the model(s), I would make following suggestions:

- 1) refer to as ependymal-like cells as equating the reprogrammed cells with bona fide ependymal cells would require a deeper molecular characterisation.
- 2) in case of mESC and radial glia, I would refer to programming rather than reprogramming, as the two starting cell populations are not terminally differentiated but ependymal cell is part of their differentiation spectrum.
- 3) With regard to reprogramming in vivo, it would be ideal if the authors could use genetic fate mapping tools to demonstrate unambiguous conversion. For example, using *Aldh1l1*ERT2 reporter mice would help demonstrating direct reprogramming of astrocytes. There is the possibility that LPS has led to downregulation of ependymal markers that become re-expressed after *Mcl*das OE. This is somewhat suggested by the fact that there are P73 positive cells still visible in controls in Fig3. Genetic fate mapping would help to clarify this.

Referee #3 (Remarks for Author):

A few points merit amendment:

Major points:

1. It does not become clear from the movies or the still in Fig 5 that it is the GFP positive cells that give rise to the cilia. Is there any possibility that the reappearance of multiciliated cells in this models is not cell autonomous? Could it be that *Mcl*das positive cells promote neo-ependymogenesis? Also, could it be that the cells have lost the plasmid as a consequence of continued proliferation? In any case, it is not clear why cells that are apparently GFP negative are multiciliated.
2. The authors might look at very early time points of the in vivo reprogramming to stain for various markers (e.g., astrocytic, radial glia etc) to better characterise the starting cells which eventually turn into ependymal-like cells.
3. The formation of pinwheels in Fig 6 requires very efficient targeting to get sufficient numbers of ependymal cells. What was the rate of electroporation? And perhaps more interesting, does the rate of successful electroporation result in a higher yield of stem cells? Finally, what is the rate of genesis of DCX positive cells in relation to electroporation success?

Finally, a question of curiosity. Is the impression correct that many induced ependymal cells seems to have a kidney shaped nucleus? If so, is this also a property of endogenous ependymal cells?

Minor points:

1. In Fig 2, the arrow points to multiple cilia in astrocytes. For an untrained eye, the red labeling looks confusing. What were the criteria to identify this red labeling with cilia?
2. The "stable" movement of the fluorescent beads does not become very clear in movie #2. It seems more movement in and out of focus.
3. Is there actually a pharmacological way to inhibit ciliar beating that could be applied to the live imaging?

Figure for reviewers removed.

(A) GemC1-knockout mouse brains were electroporated with plasmids encoding GFP or McIdas-GFP (McIdas) at postnatal day 1 (P1). Mice were sacrificed at ages P8-P11 and lateral ventricular wall whole-mount immunofluorescence was conducted with antibodies against GFP (green), and Double-cortin (DCX, red). Higher magnification of the boxed region is shown in the right panel.

(B) Analysis of the orientation of DCX⁺ migratory chains towards the ventricular wall.

We would like to thank the editor and the reviewers for their comments and suggestions, which we believe have significantly improved the quality of our manuscript.

Following the reviewers' comments, we performed additional experiments in order to:

- investigate further the potential of MclDas and GemC1 to successfully downregulate the astrocytic identity in the reprogrammed astrocytes by examining the combined expression of S100b and GFAP in transduced cortical astrocytes (Appendix Fig. S1).*
- provide evidence that reduced expression of MclDas can be a reason for the less effective reprogramming of the cortical astrocytes by GemC1 (Appendix Fig. S2).*
- characterize further ependymal cells that were generated by the ectopic expression of GemC1 or MclDas, using additional ependymal markers: Centriolin and Meig1 (Appendix Fig. S3).*
- further characterize MclDas' ability to regenerate the subventricular zone niche cytoarchitecture in our genetic hydrocephalic mouse model, by quantifying the proportion of electroporated cells that were able to form pinwheels in the V-SVZ wholemounts (Fig. 6C).*

In addition, in the revised manuscript, we have modified the presentation of graphs in all the figures.

All changes in the text are highlighted in yellow in the manuscript and a point-by-point response to reviewers' comments is attached.

*****Reviewer's comments *****

Referee #1 (Comments on Novelty/Model System for Author):

There were both in vitro (ex vivo) and in vivo models used, including primary cell cultures of early postnatal astrocytes and radial glia cells and early postnatal mice. To improve the relevance of the findings, the radial glia primary cultures should be isolated from early adult mice and selected in vivo experiments should be replicated in young adult mice.

Referee #1 (Remarks for Author):

This manuscript identifies GemC1 and MclDas, two geminin family proteins, as factors that can reprogram astrocytes and Embryonic Stem Cells (ESC) to ependymal cells. The authors use a combination of in vitro and in vivo approaches to directly cell reprogram cells to functional ependymal cells. Some of the experiments are conducted in the context of the hydrocephalus, providing a pre-clinical relevance to the data. The results are convincing and consistent and provide evidence that GemC1 and/or MclDas may potentially be used in a cell-replacement therapies of the hydrocephalus. However, the applicability of the presented results should be discussed more critically and there are important experiments required to characterize the astrocyte (or ESC) to ependyma conversion. When these experiments are added, I am confident that the manuscript would be a valuable

addition to the field of cell reprogramming and hydrocephalus.

Response to Reviewer's 1 comments:

We would like to thank the reviewer for the careful reading of the manuscript and valuable comments. Below is our response to the issues raised.

Major specific comments:

1. The segment on the statistical analysis in the methods needs to be expanded and the data must be presented correctly. The authors state that they used non-parametric statistical tests (i.e., Mann-Whitney test), however, they present all data as mean {plus minus} SEM. This is incorrect. Non-parametric tests do not compare means. Data in all graphs that were analysed by non-parametric tests should use median {plus minus} IQR or similar range indicator. The section on the statistical analysis should also indicate if the normality of data distribution was tested and if yes, why were not parametric test used. It should be made clear what statistical software was used.

Following the reviewer's comment, we have updated all our graphs in the revised version of our manuscript, to display the median {plus minus} interquartile range (IQR). Furthermore, we have replaced bar graphs with scatter plots showing individual data points. As suggested by the reviewer, we have also expanded the data analysis section in the corresponding section of Methods and provided a detailed description of the quantification as well as which normality test was used prior to the analysis.

To address the reviewer's comment, we have modified the manuscript accordingly: (Page 26-27, lines 618-624): "Quantification of overexpressed brains from was measured using the open-source platform ImageJ"

and

(Page 27, lines 625-631): "The number of independent experiments preparation were performed in GraphPad Prism 6".

2. Conversion of astrocytes to ependyma needs to be characterized in more detail. The authors use reduced or altered expression of S100b as marker of diminished astrocytic cell identity. This is not sufficient. The authors should combine S100b, GFAP and astrocytic cell morphology to demonstrate that the GemC1 or MclDas over-expressing cells lose astrocytic identity. This can be done either in vitro or in vivo.

In the original version of our manuscript, we had used GFAP and ACT co-staining in GFP or GFP-MclDas infected astrocytes and showed that there was a reduction of GFP+GFAP+ cells in MclDas-infected reprogrammed cells, suggesting a downregulation of the astrocytic cell identity. Following the reviewer's suggestion, we have now performed a more thorough characterization of the astrocytic population

assessing the combined expression of the astrocytic markers GFAP and S100 β as suggested by the reviewer, in infected astrocytes. We have quantified the number of GFAP+/S100 β + double positive cells upon GFP, GFP-MclDas and GFP-GemC1 ectopic expression. Our novel results support further our findings that MclDas is more efficient than GemC1 on reprogramming towards the ependymal lineage.

The new data have been included in the revised figure (Figure EV3A, B), and the text was modified accordingly: (Page 7, lines 143-155) "In addition we examined the ability of MclDas astrocytic identity of the transduced cells".

3. More thorough characterisation of ependymal reprogramming. The authors use P73, PCNT or FoxJ1 as markers of ependymal identity and successful reprogramming. This is not sufficient. The authors should use other ependymal markers to characterize the cell conversion more accurately and thoroughly. These markers may include Fam183b, Lrcc1 or Meig1 (MacDonald et al., Front Cell Neurosci 2021) and should be used to characterize reprogrammed astrocytes or ESC, preferably in vivo.

As the reviewer suggested, we have now employed a more detailed analysis of ependymal reprogramming using additional markers. To this end, we have performed immunofluorescence experiments with specific antibodies against the centrosomal markers Centriolin and Meig1 in transduced cortical astrocytes. Our results show that ectopic expression of MclDas to cortical astrocytes leads to accumulation of both Centriolin and Meig1, which strengthens our finding that the MclDas infected cells acquire mature ependymal cell identity.

An Appendix figure (Appendix Fig S3) has been added to the revised manuscript and the text in the revised manuscript was modified as follows: (Page 8, lines 168-173): "To address this point, we initially performed immunofluorescence experiments in transduced astrocytes verifying the generation of multiple basal bodies upon MclDas' ectopic expression".

4. Why is GemC1 less effective than MclDas in reprogramming? In all presented experiments, GemC1 is less effective in reprogramming cells to the ependymal cell fate. But why? This should be addressed. One approach could be testing how much expression of MclDas is in astrocytes, for example, where GemC1 was over-expressed.

To address the reviewer's comment, we have examined the expression levels of MclDas in cortical astrocytes following the ectopic expression of MclDas or GemC1. We have performed immunofluorescence experiments using an MclDas specific antibody in cortical astrocytes that were infected with either a GFP-MclDas or a GFP-GemC1 lentivirus and subsequently quantified MclDas fluorescence. Our analysis showed that ectopic expression of MclDas leads to increased MclDas fluorescence intensity compared to GemC1's ectopic expression during astrocytic reprogramming. Therefore, the reduced ability of GemC1 to reprogram astrocytes could be attributed

to the lower expression of *Mclidas* induced by *GemC1*'s ectopic expression, as it was pointed out by the reviewer.

In the revised version of our manuscript, an Appendix Figure (Appendix Fig.S2), that includes the new data, has been added. Additionally, we have modified the Results and Discussion sections in order to include the novel findings: (Page 7, lines 156-165) "*GemC1* is positioned upstream of *Mclidas* in the molecular pathway governing ependymogenesis, providing a potential mechanism for their differential role in ependymal reprogramming" and (Page 16, lines 347-351) "In addition, *Mclidas* overexpression was sufficient to falls within their differentiation spectrum".

5. Can astrocyte to ependyma reprogramming mitigate the hydrocephalus? In the *GemC1* KO mice, the author should determine if the *Mclidas*-driven reprogramming reduces the cross-section or volume of the lateral ventricles as a readout of resolved or reduced hydrocephalus.

We thank the reviewer for this valuable comment. We agree that it would be very interesting to observe an effect on mitigating hydrocephalus. In the experiments that we have performed we used the SVZ electroporation technique to overexpress Mclidas in the brain. The experimental procedure requires the injection of a plasmid vector encoding the protein of interest into one lateral ventricle of the mouse brain. Mice are subsequently electroporated placing a positive electrode on the dorsal lateral wall, thus only periventricular cells located on the dorsal and lateral ventricular walls of one lateral ventricle are targeted. Therefore, given the limited number of electroporated cells, which are located in one of the ventricles, in our experimental approach, it would be difficult to observe a significant reduction in the cross-section or volume of the lateral ventricles as a readout of resolved or reduced hydrocephalus.

It has been described that disruption of the ventricular/subventricular zone (V/SVZ) is a critical and consistent factor in the pathophysiology of hydrocephalus (McAllister, James P., et al. 2017, J Neuropathol Exp Neurol. 2017 May 1;76(5):358-375; Jiménez AJ et al. 2014, Tissue Barriers 2014; 2: e28426. In our study we showed that Mclidas ectopic expression restores the lost cytoarchitecture of the niche in hydrocephalic mice, suggesting that our approach could have beneficial effects on the treatment of the disease (Figure 6).

In order to clarify this point, we have adjusted the text of the Discussion section in the revised manuscript as follows: Page 18, lines 395-407: " We have used SVZ electroporation to target the periventricular cells..... human patients with hydrocephalus and ameliorating brain function.

6. The effects of *GemC1* or *Mclidas* reprogramming on the structure of V-SVZ pinwheels needs to be quantified more thoroughly. Figure 6 shows only representative images. It should show the proportion of infected (GFP+) cells that were able to form pinwheels in the V-SVZ wholemounts. In addition, the representative images of pinwheels should be shown without the cell outlines, which

obstruct

the

structures.

To address the reviewer's suggestion, we have quantified the percentage of infected cells that were able to form pinwheels in the V-SVZ wholemounts and included a new graph with our analysis in Fig. 6 of the revised manuscript. Furthermore, the representative images of pinwheels shown in Fig. 6A, B have been modified in the revised version of the manuscript by removing the cell outlines as suggested by the reviewer.

The new data have been incorporated into the text of the revised version of the manuscript as follows:

(Page 14, lines 312-314): "We also quantified the number of electroporated cells that were detected in pinwheel structures as opposed to GFP-electroporated cells that were not detected in pinwheels (Fig. 6C)".

Minor specific comments

A. Figure 3A seems to have a high background for P73 staining. Some of the P73+ cells in the panel A of LPA-McIdas appear to show background and not true P73 signal. Can a better representative image be used?

The image in panel A of LPA-McIdas has been replaced with a new representative image, following the reviewer's suggestions.

B. The representative pictures in Figure S2 are not very convincing, especially the S100b staining used to claim that the reprogrammed astrocytes change the location of S100b in the cell compartments. Better representative images should be provided.

As the reviewer suggested, the image in panel A of GemC1 in Fig. EV3 in the revised manuscript has been substituted with a new representative picture.

C. The authors should tone down the direct medical applicability of their research both in the abstract and in the discussion.

Following the reviewer's suggestion the abstract and discussion sections have been adjusted as follows:

(Page 2, lines 32-34): "Our study provides evidence future therapeutic interventions".

(Page 20, lines 437-439): "Our study shows that regeneration.... treating hydrocephalus in humans".

D. In the discussion, the authors should emphasize and highlight that their experiment were all done in early postnatal animals only and thus the applicability of their findings is limited only to postnatal and not to adult hydrocephalus.

In response to the reviewer's comment, we have revised the discussion section of our manuscript as follows:

(Page 19, lines 431-433): "Additionally, since hydrocephalus in our study appeared during early developmental stages, will require further investigation using appropriate adult animal models".

E. In the discussion, the authors state that "additional experiments in disease models closer to human pathophysiology will be required". This is a vague statement. The authors should specify what disease models closer to humans they had in mind.

The discussion section has been modified to incorporate the reviewer's suggestion. The revised text was updated as follows: (Page 19, lines 427-430): " The pig model demonstrates anatomical and physiological similarities to humans to the human conditions (Garcia-Bonilla et al, 2023)".

Referee #2 (Comments on Novelty/Model System for Author):

Fate mapping studies and deeper molecular characterization of the intermediate stages could improve a study that is nevertheless novel and well executed.

Referee #2 (Remarks for Author):

In their manuscript titled 'Ependymal cell lineage reprogramming as a therapeutic approach for hydrocephalus,' Kaplani and colleagues investigate the potential therapeutic strategy of reprogramming towards ependymal cell lineage for the treatment of hydrocephalus. The manuscript focuses on the study of GemC1 and Mclidas as 'reprogramming factors', both of which have been extensively characterized in previous studies as crucial factors in establishing the ependymal cell compartment in the adult brain by the same authors. The authors first demonstrate a role for GemC1 and Mclidas overexpression in directing the differentiation of mouse embryonic stem cells and immature astrocytes toward an ependymal cell fate in vitro. Subsequently, the authors evaluate the effects of Mclidas overexpression in vivo, where they observe a significant promotion of ependymal cell regeneration in hydrocephalus mouse models, including an ependymal cell-deficient genetic background (GemC1-KO mice). Collectively, the study introduces a novel genetic approach for inducing ependymal cell identity within resident neural cells by regulating Mclidas expression, which may hold implications for conditions related to ependymal cells, such as hydrocephalus.

The study presents compelling evidence for the role of GemC1 and, most notably, MclDas in promoting ependymal cell identity across various cellular contexts.

However, it is important to note that the term 'direct cellular reprogramming' does not precisely capture the experimental outcomes demonstrated in this study. While the experiments with mouse embryonic stem cells (mESCs) and cultured astrocytes (Fig. 1 and 2) clearly indicate the promotion of stem cell differentiation towards an ependymal fate, they do not unequivocally establish direct reprogramming. Additionally, considering the presence of neural stem cells in the ventricular wall, the in vivo experiments (Fig. 3-6) could also be interpreted as the differentiation of these progenitors towards an ependymal fate, and not as a direct change in fate from a postmitotic cell of another lineage i.e. astrocytes, as the authors indicate. Therefore, it is essential to describe the observed events more accurately, such as 'differentiation towards an ependymal fate.' It would be beneficial to address these possibilities in the discussion section, providing a more comprehensive analysis of the different potential cellular sources and differentiation mechanisms.

Response to Reviewer's 2 comments:

We would like to thank the reviewer for his suggestion. The discussion section has been modified to address the reviewer's comments. We have updated the manuscript as follows: (Page 15, lines 333-336): "We initially demonstrated that both GemC1 and MclDas overexpression in the GemC1- and MclDas overexpressing cells" and (Page 16, lines 347-351): " In addition, MclDas overexpression was sufficient to induce the ependymal fate in radial glial cells into multiciliated ependymal cells demonstrates the reprogramming potential of MclDas".

Furthermore, several experiments presented in the manuscript could potentially be interpreted as promoting the survival of ependymal cells. For instance, the in vitro experiments and the LPA model data should be considered in this context (there seems to be a large presence of GFP-negative P73 cells in the LPA model). It would be advisable to directly discuss and present this possibility as one of the potential outcomes, especially in the absence of direct fate mapping of the starting cell populations.

It has been previously demonstrated by Lummis et al. that LPA induces substantial depletion of the ependymal monolayer, while the ventricular surface becomes almost fully depleted of cilia a few hours upon LPA injection (Lummis et al. 2019, Sci Adv. 2019 Oct 9;5(10):eaax2011). They have also shown that the ependymal damage leads to apoptotic cell death in the LPA- treated mice. In accordance with this study, we show that LPA administration induces ciliary disruption alongside the ventricular walls (Fig. EV3). We also show that GFAP positive cells, which possibly correspond to reactive astrocytes, cover the walls of the lateral ventricles, as it has been previously shown in hydrocephalic mice (Páez P. et al. 200, J Neuropathol Exp Neurol. 2007 Dec;66(12):1082-927; Roales-Buján R. et al. 2012, Acta Neuropathol. 2012 Oct;124(4):531-46).

MclDas overexpression in the LPA-treated brains, targeting the lateral ventricular walls, was performed by electroporation. Since the majority of cells residing in the lateral walls of the LPA-treated mice were GFAP-positive, we hypothesized that the targeted cells would be astrocytes.

As the reviewer pointed out, immunofluorescence experiments in the electroporated LPA-treated mouse brains revealed that p73 is also expressed in GFP-negative cells. As it is also mentioned below (response to the following comment), SVZ electroporation leads to transient expression of a plasmid and therefore the loss of GFP expression can be explained by this methodological approach. Another possibility is the existence of a cell population expressing the ependymal marker p73 independently of MclDas overexpression, as it was proposed by the reviewer (see response to reviewer #3 page 11 paragraphs 3-4 as well). Consequently, we cannot exclude the possibility that electroporation might target a cell population retaining p73 expression and thus MclDas effect could be interpreted as promoting the survival of ependymal cells in this cell population. However, the significance of our results lies in successfully inducing ependymal regeneration in LPA-induced hydrocephalus independent of the population which is targeted.

In order to address the reviewer's comment, we have added the following text to the discussion section:

(Page 17, lines 374-383): "Using the LPA-induced intracranial hydrocephalus ...ependymal cells resistant to LPA-induced apoptosis".

The experiments in GemC1 knockout mice are particularly interesting and the reconstitution of functional ciliated cells and ventricular pinwheel structures elicited by MclDas overexpression is remarkable. In Figure 5 and associated movies, the authors provide evidence for cilia function. However, it appears that most of the ciliated cells are FITC/EGFP negative. If so, the authors should clarify what is the origin of those cells and otherwise provide further evidence for a direct link between the functional cells and expression of MclDas.

We previously demonstrated that GemC1-knockout mice lack ependymal cells because neural progenitor cells fail to commit to the ependymal lineage early during embryogenesis and therefore the expression of MclDas, p73, FoxJ1 were never detected in mice lacking GemC1 expression (Laloti et al. 2019, Glia. 2019, Dec;67(12):2360-2373). Therefore, the functional ciliated cells observed in GemC1-knockout mice can only be attributed to the effect of MclDas overexpression.

In addition, in the present study we overexpress MclDas by electroporating an MclDas expressing plasmid in GemC1-knockout mice. This results in transient expression of our plasmid to the targeted cell population which leads to the downregulation of expression of MclDas and GFP in longer time points. We have previously shown that during embryogenesis expression of MclDas mRNA and protein is transient and is downregulated after birth when functional specialization of the cells takes place. Our data suggest that MclDas is not required for completion of the differentiation when this process is initiated (Kyrousi et al. 2015, Development. 2015 Nov 1;142(21):3661-74; Kyrousi et al. 2017, Glia. 2017 Jul;65(7):1032-1042).

Therefore, ciliated cells that were FITC/EGFP negative might correspond to differentiating ependymal cells that have lost the expression of the MclDas transgene (see response to major point 1 of reviewer #3 as well).

The Results section of the manuscript has been modified according to the reviewer's suggestions as follows:

(Page 13, lines 285-290): "A proportion of cells with multiple beating cilia under the FITC filter.

The authors should consider/discuss the possibility that GemC1/MclDas expression alters the expression pattern of the EGFP reporter gene itself, thus biasing the studied populations, as has been reported for e.g. NeuroD1 overexpression constructs. The experiments in the ependymal-deficient mouse model are critical in excluding this possible confounding factor, but it could affect the interpretations of the other experiments.

We thank the reviewer for this valuable comment. In our experiments we have used different cellular systems and technical approaches to induce the overexpression of MclDas and GemC1 (transient transfection with plasmid vectors in mESCs, lentiviral vectors in astrocytes and electroporation with plasmid vectors in mice). Therefore, all these different approaches support the idea that the expression pattern of the GFP reporter gene does not bias the studied population. In the in vivo experiments conducted in GemC1-knockout mice in particular, plasmids expressing either internal ribosome entry site (IRES)-GFP alone or in conjunction with MclDas were utilized for overexpression of MclDas. Since IRES enables simultaneous expression of two proteins from the same bicistronic mRNA transcript, GFP and GemC1/MclDas would be produced as distinct proteins. Therefore, we hypothesize that the expression of GemC1/MclDas would not affect the expression pattern of the EGFP reporter gene itself.

Following the reviewer suggestion in order to clarify this point we have modified the Discussion section of the revised manuscript as follows:

(Page 17, lines 384-387): "Collectively, in our experimentsthe cell system and vector used".

Minor comments:

- Present replicate numbers directly in the graph as individual data points instead of only the summary bar graph

We have modified the revised version of the manuscript according to the reviewer's suggestion and all bar graphs have been replaced with scatter plots showing individual data points. In addition, graphs have been updated to display the median {plus minus} interquartile range (IQR).

- Details on how the different quantifications were performed are missing. Please explain in detail how the in vitro and particularly in vivo experiments were quantified.

To address the reviewer's comment, we have included information regarding the quantifications of both ex vivo and in vivo experiments in the Methods section. The manuscript has been elaborated as follows:

(Page 26, lines 618-624): "Quantification of overexpressed brains from in vivo experiments was performed on cryosections Fluorescent intensity per nucleus was measured using the open-source platform ImageJ.

Overall, this study presents interesting findings regarding ependymal cell regeneration in the context of hydrocephaly. Clarifying the terminology and discussing the evidence in the context of cellular differentiation will enhance the scientific rigor and completeness of the manuscript in the absence of labor-intensive fate mapping studies supporting direct reprogramming.

Referee #3 (Comments on Novelty/Model System for Author):

This manuscript by Kaplani et al. provides overall convincing evidence that forced expression of *Mcl1* and to lesser degree *GemC1* induces ependymal cell features in mouse embryonic stem cells, primary astrocytes, isolated radial glia in culture as well as presumable non-ependymal cells in vivo in pharmacological and genetic models of hydrocephalus. With regard to the model(s), I would make following suggestions:

1) refer to as ependymal-like cells as equating the reprogrammed cells with bona fide ependymal cells would require a deeper molecular characterisation.

Response to Reviewer's 3 comments:

*As suggested by the reviewer we have conducted a more detailed analysis of ependymal reprogramming using additional markers to characterize the reprogrammed cells. Specifically, we have performed immunofluorescence experiments with specific antibodies against the centrosomal markers *Centriolin* and *Meig1* in transduced cortical astrocytes. A new expanded view figure (Figure EV5) has been added to the revised manuscript further supporting our data that the reprogrammed cells can be referred as ependymal cells.*

*The text in the revised manuscript was modified as follows: (Page 8, lines 168-173): "To address this point, we initially performed immunofluorescence experiments in transduced astrocytes verifying the generation of multiple basal bodies upon *Mcl1*' ectopic expression".*

2) in case of mESC and radial glia, I would refer to programming rather than reprogramming, as the two starting cell populations are not terminally differentiated but ependymal cell is part of their differentiation spectrum.

Following the reviewer' suggestion we have modified the revised manuscript so as mESCs' and radial glial cells' conversion into ependymal stated in the text as programming instead of reprogramming.

The manuscript has been elaborated as follows:

(Page 2, lines 27-29) Our study reveals that ectopic expression of GemC1 and MclDas reprogram cortical astrocytes and program mouse embryonic stem cells into ependymal cells.

(Page 4, lines 82-85): In the present study we motile cilia in reprogrammed astrocytes.

(Page 5, lines 98-99): GemC1 and MclDas promote programming of mouse embryonic stem cells and reprogramming of cortical astrocytes into ependymal cells

(Page 5, lines 102-103): We therefore postulated that GemC1 and MclDas would be capable of inducing stem cell programming towards multiciliated ependymal cells.

(Page 5, lines 118-119): Our data show that ectopic expression of MclDas and GemC1 programs mESCs into multiciliated ependymal cells.

(Page 11, lines 247-248): We initially examined whether MclDas could program ... mice into ependymal cells ex vivo.

(Page 12, lines 260-261): These results show that cells into multiciliated ependymal cells.

3) With regard to reprogramming in vivo, it would be ideal if the authors could use genetic fate mapping tools to demonstrate unambiguous conversion. For example, using Aldh11ERT2xreporter mice would help demonstrating direct reprogramming of astrocytes. There is the possibility that LPS has led to downregulation of ependymal markers that become re-expressed after MclDas OE. This is somewhat suggested by the fact that there are P73 positive cells still visible in controls in Fig3. Genetic fate mapping would help to clarify this.

In our lab we do not have a Aldh11ERT2 reporter mice or an alternative transgenic mouse model for astrocytes' fate mapping. Conducting the suggested experiments in Aldh11ERT2 reporter mice would require us to obtain and expand this mouse line in our animal facility. However, this process would take a considerable amount of time, extending beyond the time frame for submitting the revised manuscript.

It has been previously demonstrated by Lummis et al. that LPA induces depletion of the ependymal monolayer, ependymal ciliary disruption and leads to apoptotic cell death and immune cell infiltration (Lummis et al. 2019, Sci Adv. 2019 Oct 9;5(10):eaax2011). In line with these results, we demonstrate that LPA administration induces ciliary disruption (Fig. EV3) and illustrate that GFAP-positive cells, likely representing reactive astrocytes, cover the walls of the lateral ventricles. Based on these observations, we hypothesized that MclDas overexpression via electroporation would target astrocytes.

As the reviewer pointed out, immunofluorescence experiments in the electroporated LPA-treated mouse brains revealed that p73 is also expressed in GFP-negative cells. This suggests the existence of a cell population expressing the ependymal marker p73 independently of MclDas overexpression. Consequently, we cannot exclude the possibility that electroporation might target a cell population retaining p73 expression and thus MclDas effect could be interpreted as promoting the survival of ependymal cells in this cell population. However, the significance of our results lies in

successfully inducing ependymal cell regeneration in LPA-induced hydrocephalus that could contribute to the improvement of the disease phenotype.

In order to address the reviewer's comment, we have added the following to the discussion section:

(Page 17-18, lines 374-383): "Using the LPA-induced intracranial hydrocephalus mouse model..... a subset of ependymal cells resistant to LPA-induced apoptosis".

Referee #3 (Remarks for Author):

A few points merit amendment:

Major points:

1. It does not become clear from the movies or the still in Fig 5 that it is the GFP positive cells that give rise to the cilia. Is there any possibility that the reappearance of multiciliated cells in this models is not cell autonomous? Could it be that MclDas positive cells promote neo-ependymogenesis? Also, could it be that the cells have lost the plasmid as a consequence of continued proliferation? In any case, it is not clear why cells that are apparently GFP negative are multiciliated.

We thank the reviewer for this valuable comment.

We have shown previously that GemC1-knockout mice are devoid of ependymal cells due to the inability of neural progenitor cells to commit to the ependymal lineage Lalioti et al. 2019, Glia. 2019, Dec;67(12):2360-2373. Thus, the presence of functional ciliated cells observed in GemC1-knockout mice can solely be attributed to the effect of MclDas overexpression (see also response to reviewer's #2 comment, page 8, paragraphs 3-5).

MclDas' overexpression was conducted through electroporation of a plasmid expressing MclDas, thus resulting in transient rather than stable expression of the transgene in the targeted cell population. This could provide a possible explanation for the absence of GFP signal in ciliated cells. Therefore, ciliated cells that did not express FITC/EGFP may correspond to mature ependymal cells that have lost the transgene expression.

The absence of GFP expression could not be attributed to continued proliferation since MclDas overexpression has been shown to induce cell cycle exit (Kyrousi et. al. 2015, Development. 2015 Nov 1;142(21):3661-74).

Following the reviewer's suggestions, we have adjusted the Results section of the manuscript to clarify this point:

(Page 13, lines 285-290): " A proportion of cells with multiple beating ciliaundetectable under the FITC filter.

2. The authors might look at very early time points of the in vivo reprogramming to stain for various markers (e.g., astrocytic, radial glia etc) to better characterize the starting cells which eventually turn into ependymal-like cells.

We have previously provided evidence that GemC1-deficient cells covering the walls of the lateral ventricles acquire radial glial cell characteristics and express BLBP

during early postnatal stages (an extensive characterization is described in Lalioti et al. 2019, *Glia*. 2019, Dec;67(12):2360-2373).

3. The formation of pinwheels in Fig 6 requires very efficient targeting to get sufficient numbers of ependymal cells. What was the rate of electroporation? And perhaps more interesting, does the rate of successful electroporation result in a higher yield of stem cells? Finally, what is the rate of genesis of DCX positive cells in relation to electroporation success?

We would like to thank the reviewer for this insightful comment.

We have performed additional analysis of the percentage of electroporated cells that have the ability to form pinwheel structures and have included a new graph in figure 6 (Figure 6C).

The text of the manuscript has been modified to incorporate the new data as follows:

(Page 14, lines 320-322): " We also quantified the number of electroporated cells were not detected in pinwheels (Fig. 6C)".

To address the reviewer's comment regarding the rate of genesis of DCX positive cells, whole-mount immunofluorescence was performed on ventricular walls of the lateral ventricles of GemC1-knockout hydrocephalic mice at P8-P11, previously electroporated with an MclDas-expressing plasmid or GFP alone (Figure 1-for reviewer only). We did not observe alterations in DCX+ neuroblasts' orientation towards the ventricular wall related to MclDas overexpression. Our analysis did not reveal statistically significant differences the orientation of DCX+ migratory chains between GFP and MclDas- electroporated mice, which can be attributed to the small number of cells targeted by the methodologies that we have used as it was addressed in previous comments.

Finally, a question of curiosity. Is the impression correct that many induced ependymal cells seems to have a kidney shaped nucleus? If so, is this also a property of endogenous ependymal cells?

*Ependymal cells typically exhibit a cuboidal shape (Deng et al. 2023, *Aging Dis*. 2023 Apr 1;14(2):468-483. During our analysis, we frequently observed pairs of closely located ependymal cells. We speculate that the appearance of two ependymal cells in close proximity could give the impression of a kidney shaped nucleus stated by the reviewer.*

Minor points:

1. In Fig 2, the arrow points to multiple cilia in astrocytes. For an untrained eye, the red labeling looks confusing. What were the criteria to identify this red labeling with cilia?

*Cilia are tiny hair-like structures that appear on the surface of ependymal cells (Delgehr et. 2015, *Methods Cell Biol*. 2015:127:19-35. During microscopy analysis we observed cilia as small hair-like structures nucleating from small spherical structures-the basal bodies-that appeared in different location of the z-axis compared to the nucleus in the microscope.*

Following the reviewer's comment we have modified the figure legend of Figure 2 as follows: (Page 33, lines 823-824) and we hope that it is clearer to the reader.

2. The "stable" movement of the fluorescent beads does not become very clear in movie #2. It seems more movement in and out of focus.

In live imaging experiments thick brain sections are placed onto dishes containing imaging medium with fluorescent beads. Fluorescent beads are floating on the imaging medium. Since the videos are acquired in a single plane along the z-axis minor movements can appear as shifts in and out of focus.

3. Is there actually a pharmacological way to inhibit ciliary beating that could be applied to the live imaging?

Researchers have primarily focused on identifying agents that enhance ciliary beating as a potential treatment to deal with hydrocephalus. To our knowledge, sodium metavanadate has been used as an inhibitor of ciliary movement (Nakamura & Sato, 1993, Childs Nerv Syst. 1993 Apr;9(2):65-71 Doerner et al. 2015 Elife. 2015 Dec 9;4:e11066 Previous studies on cultured ependymal cells have demonstrated that sodium metavanadate significantly reduces ciliary beating (Doerner et al). Additionally, ethanol has been reported to decrease the frequency of ciliary beating at certain concentrations. More specifically, ethanol has been shown to decrease the beating frequency of ependymal cilia in both the third and lateral ventricles of the rat brain in vivo (Omran et al. 2017, Sci Rep. 2017 Oct 20;7(1):13652).

Live imaging experiments aimed to identify ciliary beating frequency of Mclsdas reprogrammed ependymal cells in GemC1-knockout mice. Our previous research has demonstrated that GemC1-knockout mice lack ependymal cells (Laloti et al. 2019, Glia. 2019, Dec;67(12):2360-2373), thus ciliary beating is missing. A pharmacological agent to inhibit ciliary beating could serve as an additional control in our experiments.

5th Aug 2024

Dear Prof. Taraviras,

Thank you for submitting your revised study. We have now received the feedback from the 3 referees who evaluated your revised manuscript, and as you will see below, they are overall satisfied with the revisions. I am therefore pleased to inform you that I will be able to accept your manuscript once the following minor comments are addressed:

1/ Referees' concerns:

Please address the remaining concerns from referee #2 by adequate discussion and/or toning down some of the conclusions (please also consider the title of the manuscript).

2/ Manuscript text:

- Please remove the yellow highlights and only keep in track changes mode any new modification.

- Please provide up to 5 keywords.

- Methods:

o Animals: please make sure the strain and gender and of the mice are provided for all experiments. Kindly also provide the housing and husbandry conditions (food access, light/night cycle).

o Statistics: please include a statement on blinding, randomization, inclusion/exclusion criteria (and accordingly fill in the checklist section "Experimental study design and statistics").

3/ Figures and Appendix:

- Please make sure that all figures and figure panels are referenced in the text. Currently, a callout is missing for Fig. 5A.

- The correct nomenclature for the EV figure files, titles and legends should be Figure EV1, etc. (instead of Expanded View Figure 1, etc.)

- Movies: the legends should be removed from the manuscript file and each should be provided as a readme.txt file; then each movie should be zipped up together with its corresponding legend and uploaded as folder movie; the correct source file names should be Movie EV1, etc. (instead of Extended Movie 1, etc.)

- If images (or part of images) are re-used, this should be identified and indicated in the figure legends (please see Figure 5B and 5C).

- Appendix: please add page numbers, including for the table of content.

- Please address the queries of our data editors in the figure legends:

a. Please note that the exact p values are not provided in the legends of figures 1c-d, f-g; 2b-c; 3b, d-e; 4b, d-e; 5d; 6c; EV 1b-c, e; EV 2b-c, e-f; EV 4c, e; EV 5b.

b. Please indicate the statistical test used for data analysis in the legends of figures 1c-d, f-g; 2b-c; 3b, d-e; 4b, d-e; 5d; 6c; EV 1b-c, e; EV 2b-c, e-f; EV 4c, e; EV 5b.

c. Please note that scale bar and its definition are missing for figures 2d; EV 3b.

4/ Thank you for providing Source Data. Please upload them as 1 file per figure. Please also check the numerical raw data for potential unintentional duplicates (for instance figure 3D, n=2)

5/ I have introduced minor modifications in your "Paper Explained", please let me know if you agree or amend as you see fit:

Medical issue: Hydrocephalus is a condition where fluid builds up in the brain cavities, leading to serious health issues. Standard treatment involves surgery, which is risky and often fails over time. Ependymal cells have multiple tiny hair-like structures (cilia) on their surface to circulate the fluid, and their impairment may lead to hydrocephalus.

Results: We showed that GemC1 and Mclidas, which are known regulators of multiciliated ependymal cell fate determination, can push different cell types to become ependymal cells. We further managed to convert brain cells of hydrocephalic animal models into ependymal cells.

Clinical impact: Our findings suggest new therapeutic intervention aiming at the regeneration of damaged ependymal cells in human hydrocephalus.

6/ Thank you for providing a synopsis text. Could you please edit the first stand first to reflect the hypothesis / rationale of the study, and remove "out study demonstrates/our findings suggest"? You may refer to recently published articles for examples: <https://www.embopress.org/journal/17574684>

Please also suggest a striking image or visual abstract to illustrate your article as a PNG/jpeg/tiff file 550 px wide x 300-600 px high.

7/ As part of the EMBO Publications transparent editorial process initiative (see our Editorial at <http://embomolmed.embopress.org/content/2/9/329>), EMBO Molecular Medicine will publish online a Review Process File (RPF)

to accompany accepted manuscripts.

This file will be published in conjunction with your paper and will include the anonymous referee reports, your point-by-point response and all pertinent correspondence relating to the manuscript. Let us know whether you agree with the publication of the RPF. We note that you included a separate point-by-point with a figure for reviewer only. Please let us know whether this file should be excluded from the RPF.

I look forward to receiving your revised manuscript.

Yours sincerely,

Lise Roth

***** Reviewer's comments *****

Referee #1 (Remarks for Author):

The authors adequately addressed most of mine comments by new experiments or analyses. I am happy to recommend this manuscript for publication.

Referee #2 (Remarks for Author):

My previous comments, in addition to some technical notes, were centered around the appropriateness of the term direct reprogramming and the mechanisms and starting cell population underlying the remarkable 'ependymogenic' capacity of MclDas. In the revised manuscript, the authors have made some modifications to the figures and the text that have addressed some of my previous comments.

Given that the evidence used to indicate "direct cellular reprogramming" remains to be the experiments using postnatal cortical astrocyte cultures, I still find this element in the manuscript to be overemphasized and overstated. Short of in vivo fate mapping studies, which I agree are beyond the scope of the manuscript, and given that all the other evidence provided would be consistent with the targeting and differentiation of a stem/progenitor cell population, I would suggest to describe the process as differentiation towards an ependymal fate.

Likewise, I would suggest to tone down the description of the approach as therapeutic for hydrocephalus given that, as the authors explain, their in vivo experimental approach did not allow to study the functional restoration of hydrocephalus.

Minor comments:

- The experiments in GemC1 knockout mice are particularly important. Regarding the GFP-negative ciliated cells in Fig. 5, the authors mention the possibility of transient expression but then dismiss the possibility of dilution by cellular proliferation. How then would the authors explain such a transient expression if targeting a postmitotic population?

- My previous comment referred to the possibility that GemC1/MclDas expression alters the expression pattern of the EGFP reporter gene itself. To clarify, this referred to the possibility of e.g. MclDas binding and regulating the expression of the promoter element within the plasmid. In the absence of fate mapping, this has been a major confounding factor in astrocyte reprogramming (PMID 36200040). While the experiments in the ependymal-deficient mouse model (GemC1 ko) are critical in excluding this possible confounding factor, it could affect the interpretations of the other experiments and the authors should consider including it in the discussion.

Referee #3 (Comments on Novelty/Model System for Author):

The revision adequately addressed my concerns.

***** Reviewer's comments *****

Referee #2 (Remarks for Author):

My previous comments, in addition to some technical notes, were centered around the appropriateness of the term direct reprogramming and the mechanisms and starting cell population underlying the remarkable 'ependymogenic' capacity of McIdas. In the revised manuscript the authors have made some modifications to the figures and the text that have addressed some of my previous comments.

Given that the evidence used to indicate "direct cellular reprogramming" remains to be the experiments using postnatal cortical astrocyte cultures, I still find this element in the manuscript to be overemphasized and overstated. Short of in vivo fate mapping studies, which I agree are beyond the scope of the manuscript and given that all the other evidence provided would be consistent with the targeting and differentiation of a stem/progenitor cell population, I would suggest to describe the process as differentiation towards an ependymal fate.

Likewise, I would suggest to tone down the description of the approach as therapeutic for hydrocephalus given that, as the authors explain, their in vivo experimental approach did not allow to study the functional restoration of hydrocephalus.

Following the reviewer's suggestion we have modified the revised manuscript as follows:

Page 4 Lines 93-97: "Collectively, our data suggest that GEMC1 and MCIDAS can orchestrate the transcriptional program of multiciliogenesis ... potentially contribute to hydrocephalus management".

Page 15 Lines 341-344: "We initially demonstrated that both GemC1 and McIdas overexpression ... the generation of multiple cilia in the GemC1- and McIdas overexpressing cells".

Page 16 Lines 359-360: "Since this aligns within their differentiation spectrum ... differentiation into the ependymal cell lineage".

In addition, in order to tone down the description of the approach as therapeutic for hydrocephalus, as the reviewer suggested, we have made the following modifications in the revised manuscript:

Page 16 Lines 368-370 "To investigate whether McIdas would also reprogram cells in vivo... triggered by intracranial hemorrhage".

Page 19 Lines 437-439: "Together, our analysis proposes MCIDAS and GEMC1 as potent reprogramming factors ... could contribute in managing hydrocephalus".

Finally, we have altered the title of the manuscript to:

“Investigating ependymal cell lineage reprogramming as a therapeutic intervention for hydrocephalus”

Minor comments:

- The experiments in GemC1 knockout mice are particularly important. Regarding the GFP-negative ciliated cells in Fig. 5, the authors mention the possibility of transient expression but then dismiss the possibility of dilution by cellular proliferation. How then would the authors explain such a transient expression if targeting a postmitotic population?

We thank the reviewer for this valuable comment.

We have previously provided evidence that McIDAS overexpression induces cell cycle exit in McIDAS overexpressing cells (Kyrrousi et. al. 2015, Development. 2015 Nov 1;142(21):3661-74). However, we cannot exclude the possibility of continued proliferation as a possible explanation for the dilution of the expression of the GFP gene in the present study. In addition, depending on the cell cycle state of the cell which is targeted other mechanism that include plasmid degradation by host cell mechanisms (nucleases, lysosomes, endosomes, etc.) could be a possible explanation for the existence of GFP-negative ciliated cells.

Following the reviewer’s suggestions, we have adjusted the Results section of the manuscript to further clarify this point and include all the potential explanations:

Page 13, Lines 296-297: “In addition to this, we cannot ... continued proliferation”.

- My previous comment referred to the possibility that GemC1/McIDAS expression alters the expression pattern of the EGFP reporter gene itself. To clarify, this referred to the possibility of e.g. McIDAS binding and regulating the expression of the promoter element within the plasmid. In the absence of fate mapping, this has been a major confounding factor in astrocyte reprogramming (PMID 36200040). While the experiments in the ependymal-deficient mouse model (GemC1 ko) are critical in excluding this possible confounding factor, it could affect the interpretations of the other experiments and the authors should consider including it in the discussion.

We would like to thank the reviewer for this comment and the additional clarification provided on their initial feedback. Taking into consideration of the point raised by the reviewer concerning the possibility that GemC1/McIDAS expression could modify the expression pattern of the EGFP reporter gene we have modified the Discussion section in the revised manuscript as follows:

Page 18, Lines 396- 404: While our approach in the genetic mouse model of hydrocephalus ... differentiated into ependymal cells in our approach”.

8th Oct 2024

Dear Prof. Taraviras,

Thank you for bearing with the last editorial matters. I am pleased to inform you that your manuscript is accepted for publication and is now being sent to our publisher to be included in the next available issue of EMBO Molecular Medicine!

Please note that, as per our policy on competing interests (<https://www.embopress.org/competing-interests>), I have added the sentence: "Zoi Lygerou is an EMBO Council member. This has no bearing on the editorial consideration of this article for publication." in this section of the manuscript.

With kind regards,

Lise Roth
